# Quantum null energy condition and its (non)saturation in 2d CFTs

Christian Ecker[1,2⋆], Daniel Grumiller[1†], Wilke van der Schee[2,3‡],
Shahin Sheikh-Jabbari[4°] and Philipp Stanzer[1◇]

**1** Institute for Theoretical Physics, TU Wien, Wiedner Hauptstr. 8, A-1040 Vienna, Austria
**2** Institute for Theoretical Physics and Center for Extreme Matter and Emergent Phenomena,
Utrecht University, Leuvenlaan 4, 3584 CE Utrecht, The Netherlands
**3** Center for Theoretical Physics, Massachusetts Institute of Technology,
77 Massachusetts Avenue, Cambridge, MA 02139, USA
**4** School of Physics, Institute for Research in Fundamental Sciences (IPM),
P.O.Box 19395-5531, Tehran, Iran

⋆ ecker@hep.itp.tuwien.ac.at, † grumil@hep.itp.tuwien.ac.at, ‡ wilke@mit.edu,
° jabbari@theory.ipm.ac.ir, ◇ pstanzer@hep.itp.tuwien.ac.at

## Abstract

We consider the Quantum Null Energy Condition (QNEC) for holographic conformal field theories in two spacetime dimensions ($CFT_2$). We show that QNEC saturates for all states dual to vacuum solutions of $AdS_3$ Einstein gravity, including systems that are far from thermal equilibrium. If the Ryu-Takayanagi surface encounters bulk matter QNEC does not need to be saturated, whereby we give both analytical and numerical examples. In particular, for $CFT_2$ with a global quench dual to $AdS_3$-Vaidya geometries we find a curious half-saturation of QNEC for large entangling regions. We also address order one corrections from quantum backreactions of a scalar field in $AdS_3$ dual to a primary operator of dimension $h$ in a large central charge expansion and explicitly compute both, the backreacted Ryu–Takayanagi surface part and the bulk entanglement contribution to EE and QNEC. At leading order for small entangling regions the contribution from bulk EE exactly cancels the contribution from the back-reacted Ryu-Takayanagi surface, but at higher orders in the size of the region the contributions are almost equal while QNEC is not saturated. For a half-space entangling region we find that QNEC is gapped by $h/4$ in the large $h$ expansion.



# 1  Introduction

The Quantum Null Energy Condition (QNEC) [1] is a local energy condition that has attracted a lot of interest in recent years [2–10], since it can be shown to hold universally for quantum field theories in more than two dimensions (given some assumptions like unitarity) [11]. Recently, QNEC was proven assuming the averaged null energy condition [12], and the latter was proven for two-dimensional conformal field theories (CFT$_2$) in [13] (see also [14]). We are specifically interested in CFT$_2$ where the QNEC inequality takes a special form [1, 15]

$$2\pi \langle T_{\pm\pm} \rangle \geq S'' + \frac{6}{c}\left(S'\right)^2 . \tag{1.1}$$

Here $\langle T_{\pm\pm} \rangle$ are the expectation values of the null projections of the stress tensor for some state, $c$ is the central charge, $S$ is entanglement entropy (EE) for an arbitrary interval and the same state, and prime denotes variations of EE with respect to null deformations (aligned with the null direction of $T_{\pm\pm}$) of one of the endpoints of the entangling region.

It is interesting to inquire under which conditions QNEC saturates/does not saturate [7, 9, 10]. For QNEC$_2$ this issue was addressed first in [10], where it was found that QNEC saturates for the vacuum or states dual to particles on AdS$_3$ or BTZ black holes, or any state that is a Virasoro descendant of them. For the higher dimensional cases, saturation of QNEC$_4$ in systems far from thermal equilibrium was discussed in [7] and it was shown in [9] that QNEC$_d$ for $d > 2$ saturates for any local excitation.

In the present work we consider QNEC$_2$ in more detail. In most of the paper we stay in the usual AdS$_3$/CFT$_2$ context, i.e., on the bulk side we study Einstein gravity with standard asymptotic AdS$_3$ boundary conditions [16], whose solutions in absence of matter are given by the Bañados geometries [17].

We prove QNEC saturation for all states dual to the geometries using properties and solutions of Hill's equation. QNEC saturation applies not only to the global vacuum, BTZ black holes or descendants thereof, but also can apply to far from equilibrium systems. We consider as example far from equilibrium flow in quantum critical systems [18,19] and show that QNEC saturates.

Adding bulk matter makes QNEC$_2$ even more interesting, as it no longer needs to saturate. We consider both numerically and analytically examples of bulk matter to confirm the result [10] that QNEC saturates when the Ryu–Takayanagi (RT) surface does not intersect bulk matter. Specifically, we consider AdS$_3$-Vaidya geometries, which provide a gravity dual to the thermalization of a global quench on the CFT$_2$ side [20–24] and find a curious novel feature of QNEC half-saturation. We also discuss quantum corrections to QNEC from a bulk scalar field and find that QNEC is saturated to first subleading order in the central charge and the entangling interval in the small interval expansion. For the half-interval we employ additionally a large weight expansion and find that QNEC is always gapped by a quarter of the conformal weight of the scalar field.

This paper is organized as follows. In section 2 we review basic aspects of AdS$_3$/CFT$_2$, including Bañados geometries, Hill's equation and uniformization of holographic entanglement entropy (HEE). In section 3 we prove QNEC for all states dual to Bañados geometries and provide as explicit example far from equlibrium flow in quantum critical systems. In section 4 we include matter by studying AdS$_3$-Vaidya, which we study non-perturbatively numerically and perturbatively analytically, finding the phenomenon of QNEC half-saturation. In section 5 we derive quantum corrections (from a bulk scalar field) to QNEC. In section 6 we conclude with several suggestive remarks.

# 2 Basic aspects of AdS$_3$/CFT$_2$

## 2.1 Preliminaries

We consider a CFT$_2$ on a cylinder, $ds^2 = -dt^2 + d\varphi^2 = -dx^+ dx^-$ with $\varphi \sim \varphi + 2\pi$ and $x^\pm = t \pm \varphi$. Moreover, we assume that the CFT has a sparse spectrum, no gravitational anomaly and a large central charge

$$c = \frac{3}{2G_N} \gg 1 \qquad \Longleftrightarrow \qquad G_N = \text{Newton's constant} \ll 1 \qquad (2.1)$$

so that it can have an (Einstein-)gravity dual in AdS$_3$ (we set the AdS-radius to unity). The CFT$_2$ symmetries consist of two copies of the Virasoro algebra,

$$[L_n^\pm, L_m^\pm] = (n - m)L_{n+m}^\pm + \frac{c}{12}\left(n^3 - n\right)\delta_{n+m,0}. \qquad (2.2)$$

The generators $L_n^\pm$ can be viewed as Fourier modes of the (anti-)holomorphic flux components $T_{\pm\pm}(x^\pm)$ of the stress tensor.

## 2.2 Bañados geometries

On the gravity side we consider AdS$_3$ Einstein gravity with Brown–Henneaux boundary conditions. The asymptotic symmetries were found to be two copies of the Virasoro algebra

(2.2) [16].[1]  The solutions of this theory are given by the Bañados family of metrics [17],

$$ds^2 = \frac{dz^2 - dx^+ dx^-}{z^2} + \mathcal{L}^+(x^+)\left(dx^+\right)^2 + \mathcal{L}^-(x^-)\left(dx^-\right)^2 - z^2 \mathcal{L}^+(x^+)\mathcal{L}^-(x^-)\, dx^+ dx^-. \quad (2.3)$$

Note that the metric (2.3) is an exact solution to vacuum $AdS_3$ Einstein equations (and not only an asymptotic expansion around the boundary which is located at $z = 0$). The (anti-)holomorphic functions $\mathcal{L}^\pm(x^\pm)$ are assumed to be smooth and $2\pi$-periodic in their arguments. If they are positive constants we recover the family of non-extremal BTZ black holes (the extremal limit is obtained if exactly one of these constants vanishes). Poincaré patch $AdS_3$ corresponds to $\mathcal{L}^\pm = 0$ and global $AdS_3$ to $\mathcal{L}^\pm = -\frac{1}{4}$. The latter corresponds to the $SL(2, \mathbb{R})$-invariant $CFT_2$ vacuum denoted by $|0\rangle$, obeying $L_n|0\rangle = 0$ if $n \geq -1$. See [25–27] for more analysis of Bañados geometries.

$AdS_3/CFT_2$ relates the Bañados geometries (2.3) to excited $CFT_2$ states $|\mathcal{L}^+, \mathcal{L}^-\rangle$. The expectation values of the (anti-)holomorphic flux components of the stress tensor $T_{\pm\pm}(x^\pm)$ are given by the functions $\mathcal{L}^\pm$

$$2\pi \langle \mathcal{L}^+, \mathcal{L}^- | T_{\pm\pm}(x^\pm) | \mathcal{L}^+, \mathcal{L}^-\rangle = \frac{c}{6} \mathcal{L}^\pm(x^\pm). \quad (2.4)$$

Geometries with constant $\mathcal{L}^\pm$ have the same expectation values as primary states $|P\rangle$, namely $L_n|P\rangle = 0$ for positive integer $n$. Geometries with non-constant $\mathcal{L}^\pm$ can then be interpreted as Virasoro descendants of geometries with constant $\mathcal{L}^\pm$ [26].

We are interested in checking QNEC (and its possible saturation) for all these states. To this end we need to review first how to obtain HEE for states corresponding to arbitrary Bañados geometries (2.3), see [28].

## 2.3 Uniformization and Hill's equation

Since all metrics of the form (2.3) are locally $AdS_3$ there are locally (though not globally) defined diffeomorphisms mapping (2.3) to Poincaré patch $AdS_3$ [26],

$$x_P^\pm = \int \frac{dx^\pm}{\psi^{\pm 2}} - \frac{z^2 \psi^{\mp\prime}}{\psi^{\pm 2}\psi^{\mp}(1 - z^2/z_h^2)} \qquad z_P = \frac{z}{\psi^+\psi^-(1 - z^2/z_h^2)}, \quad (2.5)$$

where the functions $\psi^\pm(x^\pm)$ solve Hill's equation (to reduce clutter we suppress the arguments $x^\pm$ of all functions; prime denotes differentiation with respect to the suppressed argument)

$$\psi^{\pm\prime\prime} - \mathcal{L}^\pm \psi^\pm = 0, \quad (2.6)$$

and $z_h$ denotes the locus of one of the Killing horizons of the metric (2.3). We shall always take the outer horizon, since we are interested in applying the coordinate transformation (2.5) (which is well-defined only in a causal patch) in the asymptotic region. Denoting the two independent solutions of Hill's equation (2.6) by $\psi_{1,2}^\pm$ it is convenient to normalize to unit Wronskian

$$\psi_1^\pm \psi_2^{\pm\prime} - \psi_2^\pm \psi_1^{\pm\prime} = \pm 1, \quad (2.7)$$

which allows to express $z_h$ in terms of these solutions as $(a, b = 1, 2)$ [26]

$$z_h^2 = \frac{\psi_a^+ \psi_b^-}{\psi_a^{+\prime} \psi_b^{-\prime}}. \quad (2.8)$$

---

[1] The canonical analysis of Brown and Henneaux leads to a Poisson-bracket algebra and not commutators. We used here the canonical quantization procedure that replaces (up to a factor $i$) Poisson brackets by commutators. The same remarks apply to section 6.3 below.

## 2.4 Holographic entanglement entropy in AdS$_3$/CFT$_2$

HEE can be calculated in static configurations by computing minimal length geodesics [29] and in dynamical settings by computing geodesics (the HRT proposal [30]). These geodesics emanate from the boundary at $z = 0$ anchored at $x_1^\pm$, $x_2^\pm$. Exploiting the coordinate transformation (2.5) to the Poincaré patch of AdS$_3$ one can avoid HRT and obtain for generic Bañados metrics (2.3) the following result for HEE [28]

$$S = \frac{c}{6} \ln\left(\ell^+(x_1^+, x_2^+)\ell^-(x_1^-, x_2^-)/\epsilon^2\right), \tag{2.9}$$

with

$$\ell^\pm(x_1^\pm, x_2^\pm) = \psi_1^\pm(x_1^\pm)\psi_2^\pm(x_2^\pm) - \psi_2^\pm(x_1^\pm)\psi_1^\pm(x_2^\pm), \tag{2.10}$$

where $\psi_{1,2}^\pm$ are the appropriate solutions to Hill's equation that appear in the coordinate transformation (2.5). The result (2.9) with (2.10) is universally valid and recovers in particular all known special cases. HEE factorizes into a sum of holomorphic and anti-holomorphic contributions

$$S = S^+ + S^- \qquad S^\pm = \frac{c}{6} \ln\left(\ell^\pm(x_1^\pm, x_2^\pm)/\epsilon\right). \tag{2.11}$$

Another useful property is symmetry with respect to exchange of the anchor points, $S(x_1, x_2) = S(x_2, x_1)$. Finally, note that there is no loss of generality in translating the coordinates such that, say, $x_2^\pm = 0$, which means that effectively HEE depends only on the length and time difference between the two boundary points. For QNEC-purposes we need variations of HEE where $x_1^\pm$ varies and the other endpoint is kept fixed.

Before continuing our CFT-analysis of (H)EE let us consider the simplest example of HEE. For the Poincaré patch vacuum $\mathcal{L}^\pm = 0$ the solutions of Hill's equation (2.6) are given by $\psi_1^+ = x^+$, $\psi_2^+ = 1 = \psi_1^-$, $\psi_2^- = x^-$ so that $\ell^\pm = \pm x_1^\pm \mp x_2^\pm$, recovering [31,32]

$$S^P = \frac{c}{3} \ln \frac{\ell}{\epsilon}, \tag{2.12}$$

with $\ell = |x_1^+ - x_2^+| = |x_1^- - x_2^-|$ for a constant $t$-slice. See Eqs. (21-23) in [28] for the example of BTZ black holes.

# 3 QNEC in AdS$_3$/CFT$_2$

## 3.1 QNEC saturates for all Bañados geometries

We are now ready to provide a simple proof of QNEC saturation for all Bañados geometries (2.3), confirming the results of section 4.1 in [10]. Let us define the "vertex-operator"[2]

$$V := \exp\left(\frac{6}{c} S\right) = \frac{\ell^+(x_1^+, x_2^+)\ell^-(x_1^-, x_2^-)}{\epsilon^2} = V^+ V^- \qquad V^\pm := \frac{\ell^\pm(x_1^\pm, x_2^\pm)}{\epsilon} \tag{3.1}$$

and consider its second derivative with respect to $x_1^+$ (denoted by prime),

$$V'' = \left(\psi_1^{+\prime\prime}(x_1^+)\psi_2^+(x_2^+) - \psi_2^{+\prime\prime}(x_1^+)\psi_1^+(x_2^+)\right)\left(\psi_1^-(x_1^-)\psi_2^-(x_2^-) - \psi_2^-(x_1^-)\psi_1^-(x_2^-)\right)/\epsilon^2 = \mathcal{L}^+ V, \tag{3.2}$$

---

[2] At this stage $V$ is not actually an operator, but given the suggestive remarks in section 6.3 below it may have an operator interpretation. For our proof no such interpretation is required.

where we used the definitions (3.1) and (2.10), and in the last equality the fact that $\psi_{1,2}^+$ solve Hill's equation (2.6). Next, starting from (3.1) a simple algebra gives

$$\frac{V''}{V} = \frac{6}{c}\left(S'' + \frac{6}{c}\left(S'\right)^2\right). \tag{3.3}$$

Dividing by $V$ and multiplying by a factor $\frac{c}{6}$ the last equality (3.2) establishes QNEC saturation

$$S'' + \frac{6}{c}\left(S'\right)^2 = \frac{c}{6}\mathcal{L}^+ = 2\pi\langle T_{++}\rangle, \tag{3.4}$$

where we used the result (2.4). This concludes our proof.

QNEC saturates for all Bañados geometries for every pair of entangling interval endpoints $x_1^{\pm}, x_2^{\pm}$; while HEE depends on both of these endpoints, the special combination of HEE appearing in QNEC depends only on the coordinate with respect to which we take derivatives.

### 3.2 Example: far from equilibrium flow in quantum critical systems

The proof in the previous subsection is general; nevertheless, it is instructive to consider a nontrivial example of a Bañados geometry. For illustration we focus here on the Bañados geometry associated with a hot-cold heat bath coupling with a steady state heat current, which serves as a holographic model for far from equilibrium flow in quantum critical systems [18, 19]. A key message here is that even far from equilibrium systems can be in "quantum equilibrium" in the sense that QNEC saturates everywhere (we shall say more about this notion in section 6.2).

We take the metric (2.3) with

$$\mathcal{L}^+(x) = \mathcal{L}^-(-x) = \pi^2\left(\theta(x)\left(T_R^2 - T_L^2\right) + T_L^2\right), \tag{3.5}$$

where $\theta$ is the step function. The metric is equivalent to the one used in [19], provided one identifies $f_R(x) = f_L(x) = \mathcal{L}^+(x)/2$.

To determine HEE it is sufficient to solve Hill's equation. For the case of a step function this can be done analytically, for which we will present the solution shortly. For this example we however also plot a numerical solution, where we smoothen out the step-function, $\theta(x) = (1 + \tanh(20x))/2$. We then proceed to solve (2.6) over a sufficiently large domain, with $\psi_1^+(0) = 0, \psi_1^{+\prime}(0) = 1,\ \psi_2^+(0) = 1, \psi_2^{+\prime}(0) = 0,\ \psi_1^-(0) = 0, \psi_1^{-\prime}(0) = -1$ and $\psi_2^-(0) = 1, \psi_2^{-\prime}(0) = 0$ as boundary conditions. By our choice of the boundary conditions the solutions indeed have unit Wronskian (2.7). It is now straightforward to evaluate HEE using (2.9). Note in particular that Hill's equation has to be solved only once for a given Bañados geometry and can then be used to evaluate HEE for any boundary interval, which in particular makes it easy to evaluate the derivatives as necessary for QNEC.

The left Fig. 1 shows HEE in this geometry as a function of time for several temperature combinations for an interval that starts in the left (dashed) and right (solid) heat baths (see also [19]). From time 1 until 3 the shock wave passes the interval after which the interval is fully within the steady state heat flow, where it has temperature $\sqrt{T_L T_R}$ [18] and non-zero momentum flow. The right Fig. 1 shows the right-hand side of QNEC for a family of intervals bounded by a variable endpoint $(t_1, x_1) = (\lambda, -2 + \lambda)$ to the left and a fixed endpoint to the right at $(t_2, x_2) = (0.4, 0.2)$. For our choice of coordinates the entangling region resides on an equal time slice when $\lambda = 0.4$, for which it is of size $|x_2 - x_1| = 1.8$, and resides on a non-equal time slice for all other values of $\lambda$. Even though HEE and the stress-tensor are non-trivial for all these combinations, Hill's equation allows for an easy verification of QNEC saturation at all points, i.e. the black dashed curve $(2\pi T_{++})$ is equal to the corresponding QNEC expression.

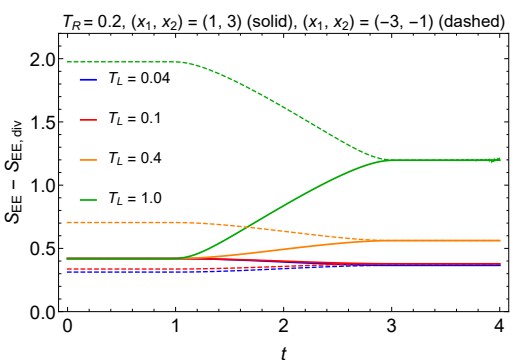
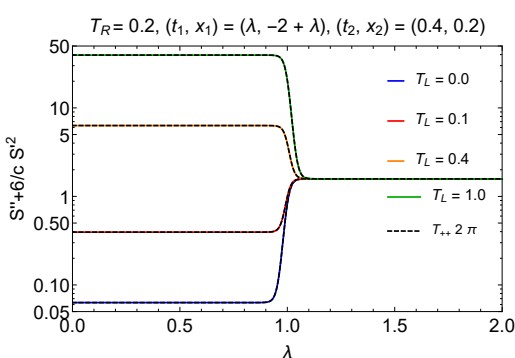

Figure 1: HEE behavior (left) and QNEC saturation (right) for far from equilibrium flow.

It is noteworthy that for the case where $\theta(x)$ above is a true step function Hill's equation

$$\psi_L'' - (\pi T_L)^2 \psi_L = 0, \quad \text{for } x < 0, \qquad \psi_R'' - (\pi T_R)^2 \psi_R = 0, \quad \text{for } x > 0, \qquad (3.6)$$

is solved by

$$\psi_{L/R}(x) = c_2 \frac{\sinh\left(\pi T_{L/R} x\right)}{\pi T_{L/R}} + c_1 \cosh\left(\pi T_{L/R} x\right), \qquad (3.7)$$

where $c_1 = \psi(0)$ and $c_2 = \psi'(0)$. From this formula it is straightforward to obtain all four solutions of Hill's equation using different combinations of $c_1$, $c_2$ and changing the sign of $x$ for the $\mathcal{L}^-$ case. These can then be used to directly evaluate HEE, $S_L$ and $S_R$, using (2.9). The four parameters of the entangling region in combination with the step function split up the final formula into twelve separate domains, all of which have a simple analytic expression for HEE, but together they are too long to reproduce here. As it must be, QNEC saturates everywhere (see appendix A for details)

$$S_L'' + \frac{6}{c}\left(S_L'\right)^2 = \frac{\pi^2 c}{6} T_L^2 \qquad\qquad S_R'' + \frac{6}{c}\left(S_R'\right)^2 = \frac{\pi^2 c}{6} T_R^2 \qquad (3.8)$$

despite of the jump in the boundary stress tensor (3.5) and the ensuing far from equilibrium flow.

# 4 QNEC in the field theory dual of AdS$_3$-Vaidya

## 4.1 Numerical approach to AdS$_3$-Vaidya

In this subsection we summarize our numerical results for HEE and QNEC in 1+1 dimensional globally quenched systems with holographic duals given by AdS$_3$-Vaidya spacetimes. In Eddington-Finkelstein coordinates the AdS$_3$-Vaidya geometry has the following line element [22, 24]

$$ds^2 = \frac{1}{z^2}\left(-(1 - M(t)z^2)\,dt^2 - 2\,dt\,dz + dx^2\right). \qquad (4.1)$$

In these coordinates the $z$-position of the apparent horizon is given by

$$z_h(t) = \frac{1}{M(t)^{1/2}}. \qquad (4.2)$$

The bulk energy momentum tensor modelling the infalling shell is given by

$$T_{tt}^{\text{bulk}}(z, t) = \frac{z}{2} M'(t). \qquad (4.3)$$

The non-vanishing components of the holographic energy momentum tensor read

$$\langle T_{tt}^{\mathrm{bdry}}(t)\rangle = \langle T_{xx}^{\mathrm{bdry}}(t)\rangle = \frac{c}{12\pi}M(t), \tag{4.4}$$

where $c$ is the central charge of the dual boundary CFT. For the profile function of the shell we choose

$$M(t) = \frac{1}{2}(1 + \tanh(at)). \tag{4.5}$$

In order to compute HEE and its derivatives relevant for the QNEC inequality we solve the non-affine geodesic equation

$$\ddot{X}^{\mu}(\sigma) + \Gamma_{\nu\rho}^{\mu}(X^{\delta}(\sigma))\dot{X}^{\nu}(\sigma)\dot{X}^{\rho}(\sigma) = J(\sigma)\dot{X}^{\mu}(\sigma) \tag{4.6}$$

subject to boundary conditions defining an entangling region of width $l$ on a constant time slice ($t = t_0$) of a cutoff surface fixed at $z = z_{\mathrm{cut}}$

$$Z(\sigma_{\pm}) = z_{\mathrm{cut}} \qquad T(\sigma_{\pm}) = t_0 \qquad X(\sigma_{\pm}) = \pm l/2, \tag{4.7}$$

where $X^{\mu}(\sigma) = (Z(\sigma), T(\sigma), X(\sigma))$ are the embedding functions of the spacelike geodesics in the ambient spacetime (4.1), $\Gamma_{\nu\rho}^{\mu}(X^{\delta}(\sigma))$ are the Christoffel symbols associated to (4.1) evaluated at the location of the geodesic and $J(\sigma)$ is the Jacobian to be defined below. We solve these equations numerically using the relaxation method explained in [33, 34]. To initialize the iterative relaxation procedure we use the following form of a pure AdS geodesic

$$Z_0(\sigma) = \frac{l}{2}(1 - \sigma^2) \qquad T_0(\sigma) = t_0 - Z_0(\sigma) \qquad X(\sigma) = \frac{l}{2}(\sigma\sqrt{2 - \sigma^2}), \tag{4.8}$$

where the Jacobian corresponding to the parameter change between the non-affine parameter $\sigma \in [\sigma_-, \sigma_+]$ and the affine parameter $\tau$, defined by $\dot{X}^2(\tau) \equiv 1$, is given by

$$J(\sigma) = \frac{\mathrm{d}^2\tau}{\mathrm{d}\sigma^2}\Big/\frac{\mathrm{d}\tau}{\mathrm{d}\sigma} = \frac{5\sigma - 3\sigma^3}{2 - 3\sigma^2 + \sigma^4}. \tag{4.9}$$

The bounds $\sigma_{\pm}$ of the non-affine parameter are chosen such that a fixed cutoff at $z = z_{\mathrm{cut}}$ is realized via

$$\sigma_{\pm} = \pm\sqrt{1 - \frac{2z_{\mathrm{cut}}}{l}}. \tag{4.10}$$

In all our numerical simulations we discretized the embedding functions with 500 grid-points and used a cutoff in $z$-direction located at $z_{\mathrm{cut}} = 0.001$. For HEE we regulate the surface areas by subtracting the corresponding vacuum results which we obtain numerically as well. As accuracy goal for the relaxation method we choose $10^{-8}$, but in most cases the residual of the finite difference equations is smaller than $10^{-10}$ already after the second iteration. In Fig. 2 we show the geodesics obtained from this procedure for the quench parameter $a = 30$. In the left plot the geodesic whose central point located at $x = 0$ touches the matter shell is highlighted in black. All geodesics with larger separation than this one cross the matter shell and have a kink-like distortion at the crossing point. The connected piece beyond the crossing point becomes a circular arc because it resides in pure AdS$_3$ where geodesics are exactly semi-circular. From the right of Fig. 2 we see that the central point (which is the endpoint of the curves) is always outside the apparent horizon. Some geodesics that cross the matter shell (colored) and have $t > z_h$ can nevertheless go beyond the apparent horizon, which then means that they have to cross the horizon four times [24]. At the special point where the central point is located at the matter shell ($t = 2.5$ in the figure) we will later see that the right-hand side of QNEC diverges.

In Fig. 3 we plot the corresponding renormalized vacuum subtracted HEE $S_{\mathrm{ren}} \equiv S - S_{\mathrm{vac}}$ as a function of the separation (left) and as a function of time (right) computed from the geodesic length.[3] Both results nicely reproduce the scaling behavior obtained from previous

---

[3]In all our plots we use the convention $G_N \equiv 1$ which is equivalent to setting the central charge $c = \frac{3}{2}$.

**Sci**|**Post**                                                            SciPost Phys. 6, 036 (2019)

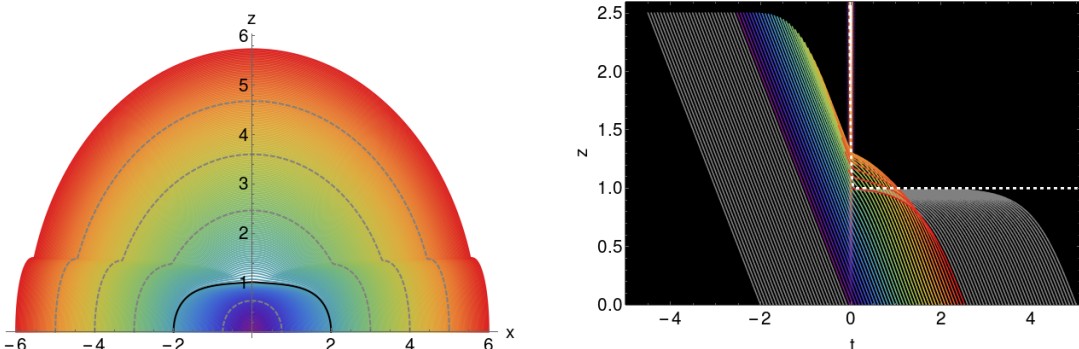

Figure 2: Left: Geodesics for different boundary separation at fixed time $t = 2$. Gray dashed lines highlight geodesics for $l = 2, 6, 8, 10$ and the solid black line is for $l = 4$ which is the separation where the central point of the geodesic, located at $x = 0$, crosses the matter shell. Right: Geodesics with fixed boundary separation $l = 5.0$ for different values of the boundary time. The white dashed line is the radial location of the apparent horizon. Colored geodesics cross the matter shell, shown as a density plot of $T_{tt}^{\text{bulk}}(z, t) = \frac{z}{2} M'(t)$ at $t \approx 0$, and do not saturate QNEC. Gray lines are geodesics that do not cross the shell and hence saturate QNEC.

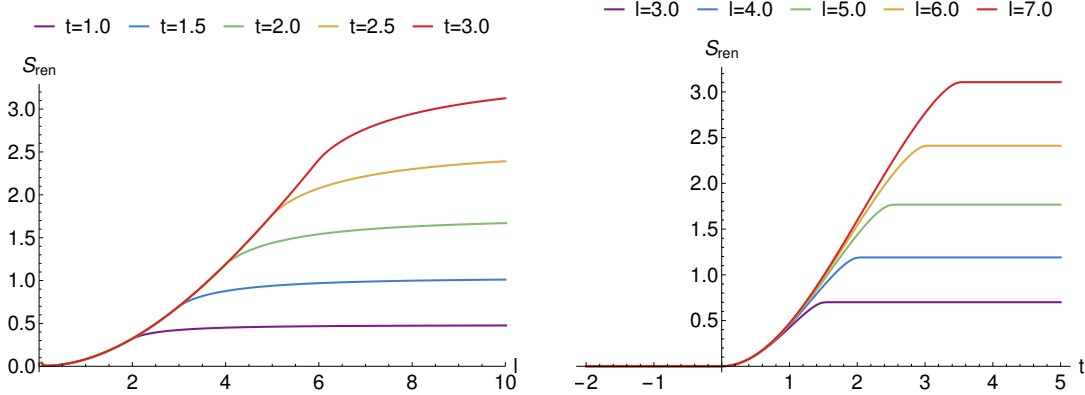

Figure 3: Left: vacuum subtracted HEE $S_{\text{ren}}$ for quench parameter $a = 30$ as function of separation $l$. Right: $S_{\text{ren}}$ as function of time.

holographic simulations [24] and the pure CFT calculations using the replica trick [35].

Let us next discuss the results for QNEC. For the computation of QNEC we compute families of seven geodesics with one endpoint shifted in light like direction $k_{p,\pm}^\mu = p\,\epsilon\,(1, \pm1)$, with $p = \{-3, -2, \dots, 3\}$ and for the shift we use $\epsilon = 0.001$. This situation is illustrated in Fig. 4, where we show such families of geodesics. From the length of these geodesics we compute the corresponding HEEs and generate a third order polynomial fit $S \approx c_0 + c_1\epsilon + c_2\epsilon^2 + c_3\epsilon^3$ from which we extract the first and second derivative at $\epsilon = 0$.

Interestingly, the outward pointing deformation $k_-^\mu$ induces only small deformations of the geodesic, where the inward pointing deformation $k_+^\mu$ induces sizable deformations even deep in the bulk. This effect is intuitive, even though it is not very apparent in the null coordinates we use. In Fefferman-Graham coordinates it would however be clearer: for $k_+^\mu$ deformation on our chosen left side the region shrinks and is shifted to later times, as opposed to $k_-^\mu$ deformations where the region grows and is shifted to later times. The shrinking entangling region means that the extremal surface also shrinks and hence probes less deep into the bulk. The later time, however, means that the infalling shell has fallen deeper into the bulk. This means

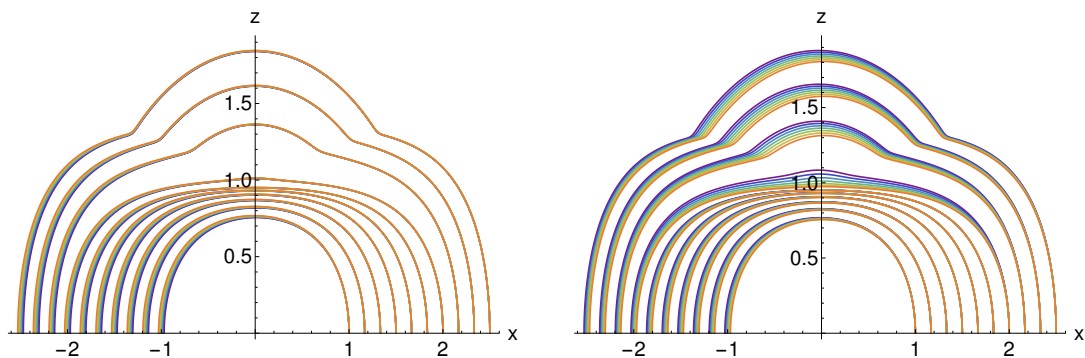

Figure 4: Families of geodesics used to compute QNEC. Colors from blue to red denote family members $p = -3$ to $p = +3$, as discussed in the main text. Left: In this plot we use a deformation vector ($k_-^\mu \propto (1, -1)$) pointing out of the entangling region. Right: This plot is for a deformation vector ($k_+^\mu \propto (1, 1)$) pointing into the entangling region. Both plots are for fixed value of the quench parameter $a = 30$ and time $t = 2.0$ and we used for illustrative purposes a rather large deformation $\epsilon = 0.01$.

that as a function of the deformation the geometry changes drastically, as is directly apparent in Fig. 4 right. For the $k_-^\mu$ direction the effect is much smaller since the extremal surface is deformed in the direction of the infalling shell. This effect is also reflected in the corresponding results for QNEC as we discuss next.

In Fig. 5 we show the right-hand side of QNEC as a function of separation for different times for negative (left) and positive (right) deformations of the entangling region. In all cases QNEC is satisfied. For separations $l < 2t$ the corresponding geodesics are too short to cross the matter shell and QNEC saturates as we demonstrated in section 3.1. For $l = 2t$ the central point of the geodesics crosses the matter shell, inducing a sharp peak in the right-hand side of QNEC for a positive deformation (this is the direction that leads to large deformations of the geodesics). The semianalytic calculation presented in the next subsection allows to analyze the features of this peak more carefully and it turns out that in the $a \to \infty$ limit, which corresponds to a $\delta$-limit of the shell, the right-hand side of QNEC diverges at this point. In case of the negative deformation the onset of non-saturation is not so violent because the geodesics are deformed along the direction of motion of the infalling shell. For $l > 2t$ QNEC cannot be saturated anymore because the geodesics always cross the matter shell. Notably, in the case of negative deformation the right-hand side of QNEC keeps on decreasing monotonically while for the positive deformation it rises again and ultimately seems to saturate at $T_{kk}/2$, as demonstrated by our perturbative analytic calculations.

We finally discuss QNEC as a function of separation $l$ and time $t$, for different values of $a$, respectively in left and right plots of Fig. 6. The left plot in Fig. 6, which shows QNEC for positive deformation $k_+$ as a function of the separation at fixed time $t = 2$ and different values of the quench parameter $a$. The peak corresponds to $l = 2t$ and it clearly becomes sharper as the shell gets thinner (which happens for larger values of $a$). As expected, QNEC is saturated for $l < 4$ and never reaches saturation for $l > 4$. In the right plot of Fig. 6 we study QNEC for positive deformation for $l = 5$ as a function of time $t$. For $t < 0$ the geodesic resides entirely in pure AdS and for $t > l/2$ in AdS-Schwarzschild; in both cases QNEC saturates. In between QNEC is not saturated because of matter shell-crossing. For increasing thickness of the shell, i.e. smaller values of $a$, the peak gets less sharp and shifted to earlier time, because the influence region of the shell extends to smaller values of $t$ which the central point of the geodesics crosses earlier. Similar logic applies to the later saturation time which is due to the

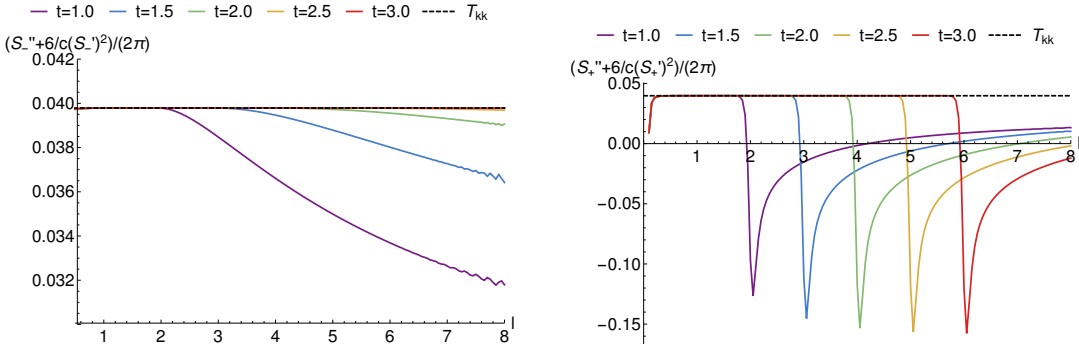

Figure 5: Left: The projected stress tensor (black dashed) $T_{kk} = \frac{1}{8\pi}M(t)$ and the right-hand side of QNEC for negative deformation ($k_-^\mu \propto (1,-1)$) as a function of separation for different boundary times. Right: QNEC for positive deformation ($k_+^\mu \propto (1,1)$). Both plots are for the quench parameter $a = 30$.

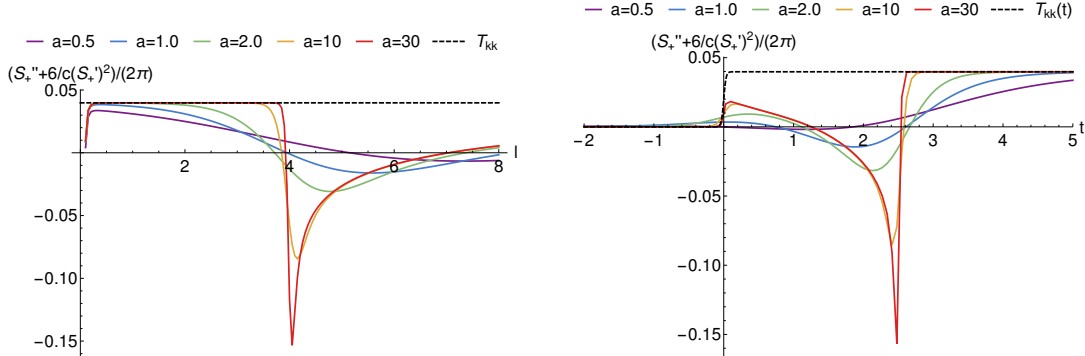

Figure 6: Right-hand side of QNEC for positive deformation ($k_+^\mu \propto (1,1)$). Left: As a function of separation for different values of $a$ and fixed boundary time $t = 2.0$. Right: As a function of time, fixed separation $l = 5.0$ and different values of $a$.

broader influence region of the matter shell which the geodesics exit only at later times. Lastly, it is interesting to note that the onset of non-saturation close to $t = 0$ approaches $T_{kk}/2$ in the large $a$ limit. We reproduce this "half-saturation" analytically in the next subsection.

## 4.2 Perturbative analysis of QNEC in AdS$_3$-Vaidya

In this subsection we study the same set-up as described in the previous one, but with a slightly more general line element of the form:

$$\mathrm{d}s^2 = \frac{1}{z^2}\left(-(1 - \epsilon M(t,z)z^2)\,\mathrm{d}t^2 - 2\,\mathrm{d}t\,\mathrm{d}z + \mathrm{d}x^2\right). \tag{4.11}$$

For this metric we then follow Ref. [10] and analytically compute QNEC for geodesics close to the vacuum result. We will see that even when the geodesics are far from the vacuum solution, it still approximates the exact numerical result remarkably well. For simplicity we restrict to equal time entangling regions.

For vacuum AdS the geodesics can be parameterized by boosting the parametrization in

[33]

$$z(\sigma) = -\frac{1}{2}\left(\sigma^2 - 1\right)\sqrt{l(2\lambda + l)}, \tag{4.12}$$

$$t(\sigma) = t_0 + \frac{1}{2}\left(\lambda(-\sigma)\sqrt{2 - \sigma^2} + \lambda + \left(\sigma^2 - 1\right)\sqrt{l(2\lambda + l)}\right), \tag{4.13}$$

$$y(\sigma) = \frac{1}{2}\sigma\sqrt{2 - \sigma^2}(\lambda + l), \tag{4.14}$$

where $\lambda$ indicates the deformation in the null direction (for brevity we take the '+' direction here, but the extension to the '-' direction is straightforward). The area integrand is then given by

$$\mathcal{A} = \int_{-1}^{1} \mathrm{d}\sigma\left(a_0(\sigma) + \epsilon a_1(\sigma)\right), \tag{4.15}$$

$$a_0 = 2\frac{1}{\sqrt{2 - \sigma^2}(1 - \sigma^2)},$$

$$a_1 = \frac{1}{4\sqrt{(2 - \sigma^2)}}\left(1 - \sigma^2\right)\left(\lambda\sigma^2 - \lambda + \sigma\sqrt{2 - \sigma^2}\sqrt{(\lambda + l)^2 - \lambda^2}\right)^2 M(t, z).$$

As the vacuum result for QNEC vanishes we can obtain the leading order term in $\epsilon$ by integrating

$$\mathcal{A}''_+ + \mathcal{A}'^2_+ = \int_{-1}^{1} \mathrm{d}\sigma\,\left(\partial_\lambda^2 a_1 + 2\partial_\lambda S_0 \partial_\lambda a_1\right), \tag{4.16}$$

where $S_0$ is the vacuum HEE, evaluated till some cut-off surface at constant $z = z_0$. The derivative of HEE is cut-off independent while the integration domain of $\sigma$ does depend on $z_0$ (though this integration domain effect is of higher order in $\epsilon$ in the integral in (4.16)).

The time derivatives of $M$ in the integrand can be eliminated by partial integration, which leads to the following integrand

$$\mathcal{A}'' + \mathcal{A}'^2 = \frac{1}{2}\left(M(t_0 - l/2, l/2) + M(t_0, 0)\right) \tag{4.17}$$

$$+ \int_{-1}^{1} \mathrm{d}\sigma\left(\frac{\sqrt{2 - \sigma^2}\left(3\sigma^6 - 7\sigma^4 + \sigma^2 - 4\sqrt{2 - \sigma^2}\sigma^3 - 1\right)\left(M(t(\sigma), z(\sigma)) - M(t_0 - l/2, l/2)\right)}{4\sigma^2}\right.$$

$$- \frac{\left(2\sigma^4 - 5\sigma^2 + 2\right)\left(-\sigma^4 + 2\sigma^2 + 2\sqrt{2 - \sigma^2}\sigma + 1\right)l\partial_z M}{4\sqrt{2 - \sigma^2}}$$

$$\left. + \frac{\sigma^2\left(\sigma^4 - 3\sigma^2 + 2\right)\left(-\sigma^4 + 2\sigma^2 + 2\sqrt{2 - \sigma^2}\sigma + 1\right)l^2\partial_z^2 M}{16\sqrt{2 - \sigma^2}}\right).$$

Care must be taken since the integrand can potentially diverge at $\sigma = 0$ after partial integration if the $\sigma = 0$ contribution is not subtracted.

As in (4.5), we take $M(t, z) = \theta(t)$, which simplifies the computation to

$$\mathcal{A}'' + \mathcal{A}'^2 = \frac{1}{2}\theta(2t_0 - l) + \frac{1}{2}\theta(t_0) + \int_{-1}^{1} \mathrm{d}\sigma\,g(\sigma)\left(\theta\left(2t_0 - (1 - \sigma^2)l\right) - \theta(2t_0 - l)\right), \tag{4.18}$$

where $g(\sigma) = \frac{1}{4}\sqrt{2-\sigma^2}\left(3\sigma^4 - 7\sigma^2 - 4\sqrt{2-\sigma^2}\sigma - 1/\sigma^2 + 1\right)$ for the plus direction and $g(\sigma) = (\sigma^2(3\sigma^6 - 11\sigma^4 + 9\sigma^2 + 8\sqrt{2-\sigma^2}\sigma - 4\sqrt{2-\sigma^2}\sigma^3 + 3))/(4(2-\sigma^2)^{3/2})$ for the minus direction. For $l < 2t_0$ the extremal surface does not cross the infalling matter shell, the integral vanishes and QNEC saturates, both for negative $t_0$ (when the stress tensor is zero), and for positive times when the metric is given by the thermal state.

$$\lim_{t_0 \gg l}\left(\mathcal{A}'' + \mathcal{A}'^2\right) = 1\,. \tag{4.19}$$

When $l \geq 2t_0$ it is possible to further simplify the integral and then perform the integration for the plus direction

$$\mathcal{A}''_+ + \mathcal{A}'^2_+ = 1 - \frac{t_0\sqrt{l^2 - 4t_0^2}(l + t_0)}{l^3} - \frac{\sqrt{l + 2t_0}}{2\sqrt{l - 2t_0}}\,, \tag{4.20}$$

and the minus direction

$$\mathcal{A}''_- + \mathcal{A}'^2_- = 1 + \frac{t_0(l - t_0)\sqrt{l^2 - 4t_0^2}}{l^3} - \frac{\sqrt{l - 2t_0}}{2\sqrt{l + 2t_0}}\,. \tag{4.21}$$

The results are shown in Fig. 7, which should be compared with the full numerical result presented in Fig. 5. The agreement is evidently excellent even at a quantitative level. This agreement is perhaps unexpected, given that the analytic calculation was done perturbatively close to the vacuum. Especially for large length and late times the geodesics in the Vaidya geometry look different from the vacuum geometry, where the latter probe much larger values of $z$. One explanation of the agreement is that we assumed an especially simple line element with $M$ independent of $z$, where the difference in shape is expected to have a smaller effect.

Several remarks are in order. First of all, it is interesting that the right-hand side of QNEC actually diverges for the positive deformation when the tip of the extremal surface at the midpoint touches the matter shell at $l = 2t_0$. Such a divergence is for instance unexpected from the point of view of (4.24) in [10], where QNEC depends on local bulk functions and the bulk stress tensor, which are all finite at the tip of the geodesics. It is also interesting that the two directions behave so differently. This is perhaps intuitive, as the plus direction deforms the extremal surface across the matter shell, and the minus direction is more constant with a deformation along the shell. Lastly, it is interesting that at least in this perturbative calculation the large $l$-limit leads to 'half-saturation' of QNEC and does not go to the thermal result, even at very late times, as long as $l$ is much larger than $t_0$. Saturation (for our units) would mean that the expressions $\mathcal{A}''_\pm + \mathcal{A}'^2_\pm$ approach 1, as they do in the limit (4.19). Instead we find from (4.20) and (4.21) half of that value in the large $l$ limit:

$$\lim_{l \gg t_0}\left(\mathcal{A}''_\pm + \mathcal{A}'^2_\pm\right) = \frac{1}{2} \mp \frac{t_0}{l} + \mathcal{O}(t_0^2/l^2)\,. \tag{4.22}$$

## 5 Finite-$c$ corrections to entanglement entropy and QNEC

Inclusion of finite-$c$ corrections in EE and the holographic computation of QNEC requires to take into account quantum corrections on the gravity side. The form of the corrections to EE was proposed in [36, 37]

$$S = \frac{\mathcal{A}}{4G_N} + \frac{\delta\mathcal{A}}{4G_N} + S_{\text{bulk}}\,, \tag{5.1}$$

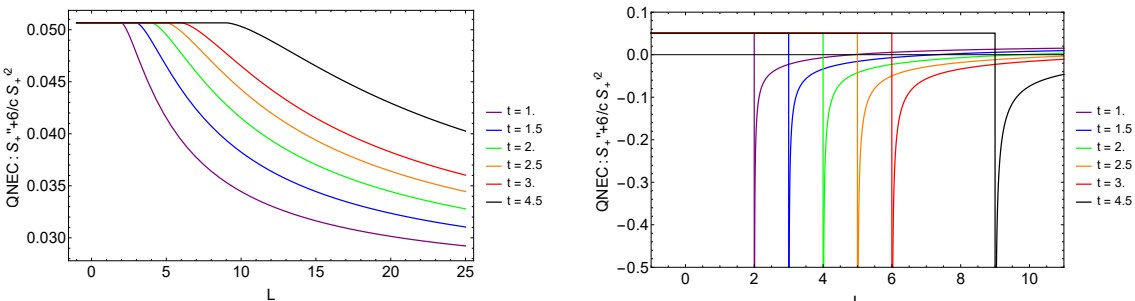

Figure 7: QNEC for perturbative Vaidya. Left: negative deformation. Right: positive deformation.

where the first term is the large-$c$ result, the second term is the change in area due to the quantum correction to the geometry and $S_{\text{bulk}}$ is the entanglement in the bulk across the extremal surface.

In this section we focus on finite-$c$ corrections to HEE and QNEC that arise when letting a massive scalar field backreact on global $\text{AdS}_3$, as pioneered in [38]. In section 5.1 we review the basic setup on gravity and field theory sides. In section 5.2 we calculate the area contribution to HEE and QNEC. In the next sections we cover the contribution of the bulk EE, which we first do perturbatively in the small interval limit in section 5.3, where we find small corrections to QNEC saturation. We also analyze another case which is complementary to the small interval case. We consider the maximal interval case, which is a half-interval region for a $\text{CFT}_2$ on a cylinder. While we have general equations on the field theory side, we can make explicit computations only when the scalar is heavy. As we show in section 5.4 the half-interval heavy scalar field case can be treated perturbatively. For this case we find that QNEC fails to saturate by a quarter of the conformal weight of the scalar perturbation.

## 5.1 Quantum backreactions from bulk scalar field

In the following we concentrate on the three-dimensional bulk setup of [38] in which a scalar field $\phi$ with mass $m^2 = 4h(h-1)$ is coupled minimally to $\text{AdS}_3$ Einstein gravity. More specifically, we consider the special case corresponding to a single scalar particle in $\text{AdS}_3$, which describes excited states in a $\text{CFT}_2$ obtained by acting with conformal primaries of weight $h$ on the vacuum state. This particle backreacts on the global $\text{AdS}_3$ geometry. In Schwarzschild-type coordinates (see below for the explicit form of the metric) the scalar field dual to a CFT primary of weight $h$ reads

$$\phi = \frac{a}{\sqrt{2\pi}} \frac{e^{-2iht}}{\left(1+r^2\right)^h} + \frac{a^\dagger}{\sqrt{2\pi}} \frac{e^{2iht}}{\left(1+r^2\right)^h}, \tag{5.2}$$

where $a$ and $a^\dagger$ are the usual annihilation and creation operators, $[a, a^\dagger] = 1$. The weight is bounded from below, $h \geq 1/2$, whereby the lowest value, $h = 1/2$, saturates the Breitenlohner–Freedman bound [39]. The bulk energy momentum tensor associated to the quantum field $\phi$ is given by

$$T_{\mu\nu} =: \partial_\mu \phi \partial_\nu \phi - \frac{1}{2} g_{\mu\nu} \big( (\nabla\phi)^2 + m^2\phi^2 \big) :, \tag{5.3}$$

where the normal ordering : ... : is chosen such that the creation operators $a^\dagger$ appear to the left of the annihilation operators $a$, which is consistent with a vanishing expectation value $\langle 0|T_{\mu\nu}|0\rangle = 0$ for the global $\text{AdS}_3$ vacuum state $|0\rangle$. For a single particle excited state generated

by $|\psi\rangle = a^\dagger|0\rangle$ the expectation value of (5.3) evaluates to [38]

$$
\begin{aligned}
\langle\psi|T_{tt}(r)|\psi\rangle &= \frac{2h(2h-1)}{\pi}\frac{1}{(1+r^2)^{2h-1}}\,,\\
\langle\psi|T_{rr}(r)|\psi\rangle &= \frac{2h}{\pi}\frac{1}{(1+r^2)^{2h+1}}\,,\\
\langle\psi|T_{\varphi\varphi}|\psi\rangle &= \frac{2hr^2}{\pi}\frac{(1-2h)r^2+1}{(1+r^2)^{2h+1}}\,.
\end{aligned}
\tag{5.4}
$$

For large values of the weight $h$ the scalar field localizes near the center $r = 0$. The leading order quantum correction (in powers of $G_N h$, which is equivalent to expanding in powers of $h/c$) to the bulk geometry follows from solving the semi-classical Einstein equations sourced by the expectation value of the stress tensor

$$
R_{\mu\nu} - \frac{1}{2}g_{\mu\nu}R - g_{\mu\nu} = 8\pi G_N \langle\psi|T_{\mu\nu}|\psi\rangle\,.
\tag{5.5}
$$

The quantum corrected geometry that solves (5.5) is known [38]

$$
\mathrm{d}s^2 = -(r^2 + G_1(r)^2)\,\mathrm{d}t^2 + \frac{\mathrm{d}r^2}{r^2 + G_2(r)^2} + r^2\,\mathrm{d}\varphi^2 \qquad \varphi \sim \varphi + 2\pi\,,
\tag{5.6}
$$

where the metric functions $G_1$ and $G_2$ are given by

$$
G_1(r) = 1 - 8G_N h + \mathcal{O}(G_N^2)
\tag{5.7}
$$

$$
G_2(r) = 1 - 8G_N h\left(1 - \frac{1}{(r^2+1)^{2h-1}}\right) + \mathcal{O}(G_N^2)\,.
\tag{5.8}
$$

For finite $G_N h$ the geometry of (5.6) is not a Bañados geometry, as it is supported by a nontrivial bulk stress tensor (5.4). Without backreactions we would have $G_1 = G_2 = 1$, in which case the metric (5.6) simplifies to global AdS$_3$.

For $h \geq 1/2$ the second term in the parentheses in (5.8) is subleading in a Fefferman–Graham expansion. Asymptotically the metric takes the form

$$
\mathrm{d}s^2 = -(r^2 + G^2)\,\mathrm{d}t^2 + \frac{\mathrm{d}r^2}{r^2 + G^2} + r^2\,\mathrm{d}\varphi^2 + \ldots \qquad G = 1 - 8G_N h\,,
\tag{5.9}
$$

which describes an AdS$_3$ geometry with conical deficit $16\pi G_N h$; it is a Bañados geometry specified by $\mathcal{L}_+ = \mathcal{L}_- = -\frac{1}{4}G^2$ (e.g. see subsection 2.2 and [25]) and the non-zero boundary stress tensor components are given by

$$
2\pi\langle T_{\pm\pm}\rangle = -\frac{c}{24}G^2 = -\frac{c}{24} + h + \mathcal{O}(h^2/c)\,.
\tag{5.10}
$$

As opposed to the conical defect solutions (5.9) the geometry (5.6) has a regular center at the origin $r \to 0$ due to backreaction by expectation values of energy momentum (5.4) associated with the scalar field.

## 5.2 RT contribution to QNEC for quantum backreacted geometry

In the following we compute the $\mathcal{A}$ and $\delta\mathcal{A}$ contributions to QNEC, first exactly and then perturbatively in the small $\Delta\varphi$ limit. In our calculations we assume $h > 1/2$, but allow for the limit $h \to 1/2$ in the end.

An extremal surface homologous to an interval $(t_1, \varphi_1) = (0,0)$, $(t_2, \varphi_2) = (\lambda, \Delta\varphi + \lambda)$ at the boundary $z = 0$ can be represented as $z(\varphi)$ and $t(\varphi)$, such that the relevant area functional takes the form

$$\mathcal{A} = \int_0^{\Delta\varphi+\lambda} d\varphi \, \mathcal{L}(z, \dot{z}, \dot{t}) = \int_0^{\Delta\varphi+\lambda} d\varphi \, \frac{1}{z} \sqrt{1 + \frac{\dot{z}^2}{f_2(z)} - \dot{t}^2 f_1(z)}, \qquad (5.11)$$

where $\lambda$ parametrizes a small deformation of the entangling region and $f_{1,2}$ are given by

$$f_1(z) = 1 + z^2 \left(1 - 8G_N h\right)^2, \qquad (5.12)$$

$$f_2(z) = 1 + z^2 \left[1 - 8G_N h\left(1 - \left(1 + z^{-2}\right)^{1-2h}\right)\right]^2. \qquad (5.13)$$

Since we need only terms up to second order in $\lambda$ it is useful to expand to this order before performing calculations, and we do this at every step. The area functional (5.11) is invariant under $\varphi \to \varphi + \delta\varphi$ yielding the Noether charge

$$Q_1 = \mathcal{L} - \dot{z}\frac{\partial\mathcal{L}}{\partial\dot{z}} - \dot{t}\frac{\partial\mathcal{L}}{\partial\dot{t}} = \frac{1}{z\sqrt{1 + \dot{z}^2/f_2(z) - \dot{t}^2 f_1(z)}} =: \frac{1}{z_* N_*}. \qquad (5.14)$$

We express the constant of motion $N_*$ using the $z$-coordinate of the bulk geodesic $z_* := z(\varphi_*)$ at its maximal $z$-value where $\dot{z}(\varphi_*) = 0$

$$N_* = \sqrt{1 - \left(\dot{t}^2 f_1(z)\right)\big|_{z=z_*}}. \qquad (5.15)$$

There is a second Noether charge associated to time translation invariance

$$Q_2 = \partial\mathcal{L}/\partial\dot{t} = -\frac{\dot{t} f_1(z)}{z\sqrt{1 + \dot{z}^2/f_2(z) - \dot{t}^2 f_1(z)}}. \qquad (5.16)$$

Dividing the two Noether charges $Q_{1,2}$ gives a second constant of motion

$$\Lambda := -\frac{Q_2}{Q_1} = \dot{t} f_1(z). \qquad (5.17)$$

Combining (5.14), (5.15) and (5.17) yields

$$\dot{z}_\pm = \pm\sqrt{(\Lambda^2/f_1(z) + N_*^2 z_*^2/z^2 - 1)f_2(z)}, \qquad (5.18)$$

where the positive branch $\dot{z}_+$ corresponds to the interval $0 \le \varphi \le \varphi_*$ and the negative branch $\dot{z}_-$ to $\varphi_* \le \varphi \le \Delta\varphi + \lambda$. The boundary conditions of the extremal surface fix the values of the Noether charges, so that $\Delta\varphi + \lambda$ can be expressed using (5.18),

$$\Delta\varphi + \lambda = \int_0^{\Delta\varphi+\lambda} d\varphi = 2\int_0^{z_*} \frac{dz_+}{\dot{z}_+} = 2z_*\int_0^1 dx \, \frac{x}{R(x)}\sqrt{\frac{f_1(z_* x)}{f_2(z_* x)}}, \qquad (5.19)$$

where we switched to the dimensionless variable $x = z_+/z_*$ and defined

$$R(x) := \sqrt{\Lambda^2 x^2 + f_1(z_* x)(N_*^2 - x^2)}. \qquad (5.20)$$

Similarly, the integral for the shift in $t$-direction

$$\lambda = \int_0^\lambda dt = 2\int_0^{z_*} dz_+ \frac{\dot{t}}{\dot{z}_+} = 2\Lambda z_*\int_0^1 dx \, \frac{x}{R(x)\sqrt{f_1(z_* x)f_2(z_* x)}} \qquad (5.21)$$

yields

$$\lambda = 2\arctan\left(\frac{\Lambda z_*}{\sqrt{(1+z_*^2)(1+z_*^2-\Lambda^2)}}\right) + \frac{32 G_N h \Lambda z_*^3}{(1+z_*^2)^2} - \frac{8 G_N h \Lambda z_* Z}{(1+z_*^2)} + \mathcal{O}(G_N^2) \tag{5.22}$$

with the definition

$$Z := z_*^{4h}(1+z_*^2)^{-2h}\frac{\sqrt{\pi}\,\Gamma[2h+1]}{\Gamma[2h+\frac{3}{2}]}. \tag{5.23}$$

As a next step we expand the right-hand side of (5.22) to $\mathcal{O}(\Lambda^2)$ and solve for $\Lambda$

$$\Lambda = \frac{\lambda}{2z_*}\left(1+z_*^2-16 G_N h z_*^2+4 G_N h(1+z_*^2)Z\right). \tag{5.24}$$

Inserting this expression into (5.19), expanding in $\lambda$ and integration yields

$$\Delta\varphi + \lambda = 2(1+8 G_N h)\arctan z_* + \frac{\lambda^2}{4z_*} - \frac{16 G_N h z_*}{1+z_*^2} - 8 G_N h z_* \,_2F_1(\tfrac{1}{2},1,2h+\tfrac{3}{2};-z_*^2)\,Z + \frac{G_N h \lambda^2}{z_*}Z, \tag{5.25}$$

which allows to express $z_*$ in terms of $\Delta\varphi$ and $\lambda$. While we have the exact expressions, they are somewhat lengthy, so we display in the paper only the leading order version exactly and the subleading order perturbatively in a small $\Delta\varphi$-expansion:[4]

$$z_* = \tan\frac{\Delta\varphi}{2} + \frac{\lambda}{2}\frac{1}{\cos^2\frac{\Delta\varphi}{2}} + \frac{\lambda^2}{4}\left(\frac{\tan\frac{\Delta\varphi}{2}}{\cos^2\frac{\Delta\varphi}{2}} - \frac{1}{\sin\Delta\varphi}\right) + \mathcal{O}(G_N). \tag{5.26}$$

Our last task is to evaluate the area functional (5.11), going away from the boundary $z=0$ to a cut-off surface at $z=z_{\text{cut}}$. Inserting into this functional the expressions for $\Lambda$ and $z_*$ above yields

$$\mathcal{A} = 2\int_{z_{\text{cut}}/z_*}^{1} dx\, \frac{\sqrt{1-\Lambda^2/f_1(z_*)}}{xR(x)}\sqrt{\frac{f_1(z_* x)}{f_2(z_* x)}} = 2\ln\frac{z_*}{z_{\text{cut}}} + \int_0^1 dx\, I_\mathcal{A}^{(0)} + \int_0^1 dx\, I_\mathcal{A}^{(1)} + \mathcal{O}(\lambda^3) \tag{5.27}$$

with the integrand

$$I_\mathcal{A}^{(0)} = \frac{2(S(x)-1)}{x} - \frac{2\lambda S(x)x\tan\frac{\Delta\varphi}{2}}{1+x^2+(1-x^2)\cos\Delta\varphi} + \frac{\lambda^2 S(x)x\big((1-x^2)\cos\Delta\varphi-2+x^2\big)}{\big(1+x^2+(1-x^2)\cos\Delta\varphi\big)^2}, \tag{5.28}$$

where

$$S(x) := \left(1-x^2\right)^{-1/2}\left(1+x^2\tan^2\frac{\Delta\varphi}{2}\right)^{-1/2} \tag{5.29}$$

and the order $G_N$-term $I_\mathcal{A}^{(1)}$ is again too long to be displayed here.

The limit $\lambda \to 0$ of (5.27) determines the RT contribution at an equal time slice, which after symmetrizing with respect to $\Delta\varphi \leftrightarrow 2\pi - \Delta\varphi$ yields

$$S_{\text{RT}} = \frac{1}{4G_N}\mathcal{A}\Big|_{\lambda\to 0} = \frac{c}{3}\ln\frac{2\sin\frac{\Delta\varphi}{2}}{z_{\text{cut}}} + 4h\left(1 - \frac{\pi-|\pi-\Delta\varphi|}{2}\cot\frac{\pi-|\pi-\Delta\varphi|}{2}\right.$$
$$\left. + \frac{\sqrt{\pi}\,\Gamma[2h]}{\Gamma[2h+\frac{3}{2}]}\sin^{4h}\frac{\Delta\varphi}{2}\left(h\,_2F_1(\tfrac{1}{2},1,2h+\tfrac{3}{2};-\tan^2\frac{\Delta\varphi}{2})-h-\frac{1}{4}\right)\right) + \mathcal{O}(1/c). \tag{5.30}$$

---

[4]Terms of order $G_N$ are needed as well; their explicit form and all the details of the quantum corrected QNEC calculation are available on the webpage http://quark.itp.tuwien.ac.at/~grumil/QNEC_quantum_correct.nb; in our final results for HEE and QNEC we display also the first subleading terms in the large-$c$ expansion.

In the small interval expansion the expression above simplifies to

$$S_{\text{RT}} = \frac{c}{3} \ln \frac{\Delta\varphi}{z_{\text{cut}}} - \frac{c\Delta\varphi^2}{72} + \frac{h\Delta\varphi^2}{3} + \mathcal{O}(\Delta\varphi^4) + \mathcal{O}(1/c). \tag{5.31}$$

The results (5.30) and (5.31) agree precisely with the ones in [38] [see their Eqs. (4.11) and (4.12)].

First and second derivatives of (5.27) with respect to $2\lambda$ give the right-hand side of the QNEC inequality (1.1) [the reason we use $2\lambda$ rather than $\lambda$ stems from our conventions for light-cone coordinates so that the stress tensor is normalized as in (5.10)]. Keeping all orders in $\Delta\varphi$ after a lengthy but straightforward calculation we find the exact expression (valid for positive integer or half-integer weights $h$)

$$\text{RT part:} \qquad S_{\text{RT}}'' + \frac{6}{c}\left(S_{\text{RT}}'\right)^2 = -\frac{c}{24} + h - \frac{h\sqrt{\pi}\,\Gamma[2h+2]}{4\Gamma[2h+\frac{3}{2}]} \sin^{4h-2}\frac{\Delta\varphi}{2} + \mathcal{O}(1/c). \tag{5.32}$$

The result above gives the RT contribution to QNEC. The full expression for QNEC also involves the bulk EE which we compute in the next two subsections for two different scenarios. We start with the small interval limit.

## 5.3 QNEC contribution of bulk entanglement entropy for small interval

A convenient way to estimate the difference of the EE of a small perturbation of the vacuum and the vacuum state is to compute the expectation value of the modular Hamiltonian

$$\Delta S = 2\pi\,\Delta\langle H_0 \rangle, \tag{5.33}$$

where $H_0$ is the modular Hamiltonian of the vacuum (given by $\rho_{\text{vac}} = e^{-2\pi H_0}$). One way to ensure that the state is indeed a small perturbation is to take the small interval limit, which is the regime of applicability of this subsection. We are interested in $\lambda$-deformed regions of EE, which goes beyond the case considered in [38], where only equal-time entangling regions were computed. It is in principle straightforward to extend this analysis by going to a boosted frame

$$\begin{pmatrix} t' \\ \varphi' \end{pmatrix} = \begin{pmatrix} 1+\zeta^2/2+\mathcal{O}(\zeta^4) & \zeta+\mathcal{O}(\zeta^3) \\ \zeta+\mathcal{O}(\zeta^3) & 1+\zeta^2/2+\mathcal{O}(\zeta^4) \end{pmatrix} \begin{pmatrix} t \\ \varphi \end{pmatrix}, \tag{5.34}$$

where rapidity is given by $\zeta = -\epsilon - \epsilon^2 + \mathcal{O}(\epsilon^3)$ with $\epsilon = \lambda/\Delta\varphi$ (the boosted frame has velocity $\delta\varphi/\delta t = \lambda/(\Delta\varphi+\lambda) \approx \epsilon + \epsilon^2$). After boosting, again HEE can be computed on an equal time slice. The expectation value of the modular Hamiltonian is then given by [38, 40]

$$\Delta\langle H_0 \rangle = \int_{\Sigma_A} d\Sigma_A \sqrt{|g_{\Sigma_A}|}\, \xi^\nu n^\mu \langle\psi|T_{\mu\nu}|\psi\rangle, \tag{5.35}$$

where $g_{\Sigma_A}$ is the induced metric on the entanglement wedge $\Sigma_A$, $\xi^\nu = (1,0,0)$ is the Killing vector generating Rindler-time translations, and $n^\mu = ((\rho^2-1)^{-1/2},0,0)$ is the normal vector to $\Sigma_A$ (all in Rindler coordinates, see [38] for the explicit transformation to apply to the boosted $T_{\mu\nu}$).

The integral can be performed numerically, but in order to solve (5.35) analytically we need a small parameter, for which we chose a small $\Delta\varphi$ expansion. The results for this small length expansion are summarized in Table 1 and Fig. 8. For $h > 1/2$ QNEC is satisfied for all $\Delta\varphi < \pi$ and saturates in the limit $\Delta\varphi \to 0$. For any positive (half-)integer value of $h$ the leading order terms $\Delta\varphi^{4h-2}$ in the RT-contribution cancel precisely with the ones coming from bulk entanglement, reminiscent of a similar cancellation between RT- and bulk contributions

Table 1: All leading terms in a small $\Delta\varphi$ expansion cancel

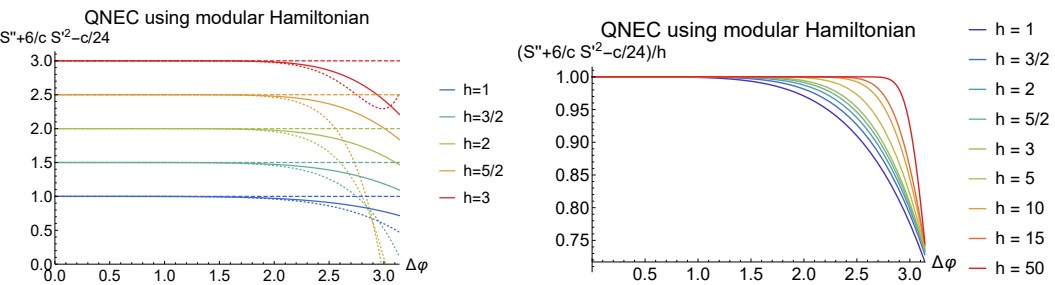

| $h$ | $\Delta\langle H_0\rangle'' + \frac{12}{c}\Delta\langle H_0\rangle' S_0'$ | $S_{\mathrm{RT}}'' + \frac{6}{c}(S_{\mathrm{RT}}')^2 - h + \frac{c}{24}$ |
|---|---|---|
| 1/2 | $\frac{1}{3} + \mathcal{O}\left(\Delta\varphi^2\right)$ | $-\frac{1}{3}$ |
| 1 | $\frac{\Delta\varphi^2}{5} - \frac{43\Delta\varphi^4}{2520} + \frac{\Delta\varphi^6}{21600} + \mathcal{O}\left(\Delta\varphi^8\right)$ | $-\frac{\Delta\varphi^2}{5} + \frac{\Delta\varphi^4}{60} - \frac{\Delta\varphi^6}{1800} + \mathcal{O}\left(\Delta\varphi^8\right)$ |
| 3/2 | $\frac{3\Delta\varphi^4}{35} - \frac{73\Delta\varphi^6}{5040} + \frac{1571\Delta\varphi^8}{1663200} + \mathcal{O}\left(\Delta\varphi^{10}\right)$ | $-\frac{3\Delta\varphi^4}{35} + \frac{\Delta\varphi^6}{70} - \frac{3\Delta\varphi^8}{2800} + \mathcal{O}\left(\Delta\varphi^{10}\right)$ |
| 2 | $\frac{2\Delta\varphi^6}{63} - \frac{667\Delta\varphi^8}{83160} + \frac{2789\Delta\varphi^{10}}{3088800} + \mathcal{O}\left(\Delta\varphi^{12}\right)$ | $-\frac{2\Delta\varphi^6}{63} + \frac{\Delta\varphi^8}{126} - \frac{\Delta\varphi^{10}}{1080} + \mathcal{O}\left(\Delta\varphi^{12}\right)$ |
| 5/2 | $\frac{5\Delta\varphi^8}{462} - \frac{11\Delta\varphi^{10}}{3024} + \frac{7387\Delta\varphi^{12}}{12972960} + \mathcal{O}\left(\Delta\varphi^{14}\right)$ | $-\frac{5\Delta\varphi^8}{462} + \frac{5\Delta\varphi^{10}}{1386} - \frac{19\Delta\varphi^{12}}{33264} + \mathcal{O}\left(\Delta\varphi^{14}\right)$ |
| 3 | $\frac{\Delta\varphi^{10}}{286} - \frac{151\Delta\varphi^{12}}{102960} + \frac{3907\Delta\varphi^{14}}{13366080} + \mathcal{O}\left(\Delta\varphi^{16}\right)$ | $-\frac{\Delta\varphi^{10}}{286} + \frac{5\Delta\varphi^{12}}{3432} - \frac{\Delta\varphi^{14}}{3432} + \mathcal{O}\left(\Delta\varphi^{16}\right)$ |

Figure 8: Finite-$c$ corrected QNEC. Left: The solid lines show the right-hand side of QNEC which includes the modular Hamiltonian and the RT extremal surface contributions to the EE. The dotted lines show the small $\Delta\varphi$ expansion as computed in Table 1. Dashed lines represent the corresponding values of $2\pi(\langle T_{\pm\pm}\rangle - c/24)$. Right: QNEC non-saturation, rescaled by $h$. As we see QNEC non-saturation happens at large intervals. For large $h$ the non-saturation appears only for intervals close to the half-space ($\Delta\varphi = \pi$). We derive in Section 5.4 the result suggested by the right plot, namely that QNEC is gapped by $h/4$ at $\Delta\varphi = \pi$ for large $h$.

to HEE observed in [38]. However, we do not have a cancellation beyond these leading order terms. Our main result for QNEC at small interval is

$$\text{Small interval:} \qquad 2\pi\langle T_{\pm\pm}\rangle - S'' - \frac{6}{c}\left(S'\right)^2 = +\mathcal{O}(\Delta\varphi^{4h}). \qquad (5.36)$$

Here $S = S_{\mathrm{RT}} + S_{\mathrm{bulk}}$ contains the full RT-contribution and the leading order bulk corrections. The plus sign on the right-hand side indicates that the first correction always is positive, so that QNEC always holds.

Importantly, as in [38] the modular Hamiltonian only approximates HEE up to $\mathcal{O}((\Delta\varphi)^{8h-1})$, after which it is possible to compute HEE directly using a Bogoliubov transformation.

## 5.4 QNEC contribution of bulk entanglement entropy at large weight

### 5.4.1 General remarks on large weight limit and half-interval

There is another simple way to have a small parameter, namely to consider the double limit $c \gg h \gg 1$. The first inequality guarantees that backreactions remain small, while the second one introduces $1/h$ as small parameter. From previous results, as depicted in Fig. 8 we expect that QNEC saturates at large $h$ up to tiny corrections for any interval $\Delta\varphi < \pi$. However, this ceases to be true for the half-interval, $\Delta\varphi = \pi$. In this subsection we consider the half-interval in the large weight limit, which allows us to evaluate various integrals using the saddle point approximation.

The special case where the entangling region is half the circle, $\Delta\varphi = \pi$, is the only one where we can expect non-trivial corrections to EE for arbitrarily large weights $h$. As we shall see, evaluating QNEC for this case leads to an interesting result. We shall say more about the validity of this approach in the concluding section 6.1; for now we simply assume that the half-interval case can be computed at large $h$ and proceed with the calculations.

### 5.4.2 Holographic entanglement entropy for half-interval

The RT-corrected result for HEE (5.30) at $\Delta\varphi = \pi$ and large $c$ and $h$ simplifies to

$$S_{\text{RT}} = \frac{c}{3}\ln\frac{2}{z_{\text{cut}}} + 4h - \frac{\sqrt{\pi}\Gamma(2h+1)}{\Gamma\left(2h+\frac{1}{2}\right)} = \frac{c}{3}\ln\frac{2}{z_{\text{cut}}} + 4h - \sqrt{2\pi h} + \mathcal{O}(1/c) + \mathcal{O}(1/\sqrt{h}). \quad (5.37)$$

For EE at half-interval the integral of the modular Hamiltonian (5.35) becomes surprisingly simple,

$$\Delta\langle H_0\rangle = 4h(2h-1)\int_1^\infty d\rho \int_{-\infty}^\infty dx\, \rho\,(\rho\cosh(x))^{-4h}. \quad (5.38)$$

This integral evaluates to

$$\Delta\langle H_0\rangle = \frac{\sqrt{\pi}\Gamma(2h+1)}{\Gamma\left(2h+\frac{1}{2}\right)} = \sqrt{2\pi h} + \mathcal{O}(1/\sqrt{h}). \quad (5.39)$$

Adding (5.39) to (5.37), the total order one correction in the large $c$ expansion to EE is given by $4h$.

Alternatively, it is possible to compute the bulk corrections to HEE using the Bogoliubov coefficients calculated in [38] [see their Eq. (3.31)]. For $\Delta\varphi = \pi$ the Bogoliubov coefficients read

$$\alpha = (-i)^{i\omega}F \qquad \beta = -i^{i\omega}F, \quad (5.40)$$

where $F$ is defined as in (3.32) of [38]. This implies

$$|\alpha|^2 = e^{\pi\omega}|F|^2 \qquad |\beta|^2 = e^{-\pi\omega}|F|^2 \quad (5.41)$$

and insertion into the bulk Rindler modular Hamiltonian[5] $H_R$

$$2\pi\Delta\langle H_R\rangle = \int\frac{d\omega}{2\pi}\int\frac{dk}{2\pi}\,2\pi\omega\left(|\alpha|^2 + |\beta|^2\right) \quad (5.42)$$

yields

$$2\pi\Delta\langle H_R\rangle = \int d\omega \int dk\, \frac{2^{4h}\,\omega\coth(\pi\omega)}{4\pi\,(\Gamma[2h])^2}\prod_{\pm,\pm}\Gamma\Big[h\pm i\frac{k\pm\omega}{2}\Big], \quad (5.43)$$

where the product goes over all four combinations of signs.

To evaluate the integrals we assume from now on large weight $h \gg 1$ and exploit the saddle point approximation

$$\lim_{h\to\infty}\int_{-\infty}^\infty dk\, e^{-hf(k)} \approx \lim_{h\to\infty}\sqrt{\frac{2\pi}{h f''(k_s)}}\,e^{-hf(k_s)}, \quad (5.44)$$

---

[5]The calculation using Bogoliubov coefficients has the advantage of being generalizable to higher orders in the excitation density matrix, but here we confine ourselves to a first order calculation, so that $2\pi\Delta\langle H_R\rangle$ calculated here should reduce to $\Delta\langle H_0\rangle$ calculated above. We shall see that this is indeed the case.

where $k_s$ is the value of $k$ that extremizes $f(k)$, i.e., $f'(k_s) = 0$, assuming that there is exactly one such value. Together with Stirling's formula

$$\lim_{z \to \infty} \ln \Gamma[z+1] = z\big(\ln(z) - 1\big) + \frac{1}{2} \ln(2\pi z) + \mathcal{O}(1/z) \tag{5.45}$$

and rescalings $k = h\tilde{k}$, $\omega = h\tilde{\omega}$ the integral (5.43) can be rewritten as

$$2\pi \Delta \langle H_R \rangle = \int d\tilde{\omega} \, \frac{2^{4h} h^3 \tilde{\omega} \coth(\pi h \tilde{\omega})}{4\pi \, (\Gamma(2h))^2} \int d\tilde{k} \, e^{-hf(\tilde{k}, \tilde{\omega})} \tag{5.46}$$

with

$$f(\tilde{k}) = 4 + \big(\tilde{k} + \tilde{\omega}\big) \arctan \frac{\tilde{k} + \tilde{\omega}}{2} + \big(\tilde{k} - \tilde{\omega}\big) \arctan \frac{\tilde{k} - \tilde{\omega}}{2} - 4 \ln \frac{h}{2} - \ln \Big( \big(4 + (\tilde{k} + \tilde{\omega})^2\big)\big(4 + (\tilde{k} - \tilde{\omega})^2\big)\Big)$$
$$+ \frac{2}{h} \ln \frac{h}{2\pi} + \frac{1}{2h} \ln \Big( \big(1 + (\tilde{k} + \tilde{\omega})^2/4\big)\big(1 + (\tilde{k} - \tilde{\omega})^2/4\big)\Big) + o(1/h). \tag{5.47}$$

The stationary point that extremizes the function $f$ in (5.47) is located at $\tilde{k} = 0$ (by plotting the function for some sample values of $\tilde{\omega}$ one can check that this is the only extremum of the function). The second derivative at the extremum evaluates to

$$f''(\tilde{k} = 0, \tilde{\omega}) = \frac{1}{1 + \tilde{\omega}^2/4} - \frac{2}{h} \frac{\tilde{\omega}^2 - 4}{\tilde{\omega}^2 + 4} + \mathcal{O}(1/h^2), \tag{5.48}$$

while the function itself yields

$$f(\tilde{k} = 0, \tilde{\omega}) = 4 - 4 \ln \frac{h}{2} - (2 - i\tilde{\omega}) \ln(2 - i\tilde{\omega}) - (2 + i\tilde{\omega}) \ln(2 + i\tilde{\omega}) + \frac{1}{h} \ln \frac{h^2 \big(\tilde{\omega}^2 + 4\big)}{16\pi^2} + o(1/h). \tag{5.49}$$

Inserting (5.44) (with $\tilde{k}_s = 0$) and (5.47) into (5.46) yields

$$2\pi \Delta \langle H_R \rangle = \sqrt{2\pi h} \int d\hat{\omega} \, \hat{\omega} \coth(\pi \sqrt{h} \hat{\omega}) \left(1 + \frac{\hat{\omega}^2}{4h}\right)^{2h - 1/2} \left(\frac{2 + i\hat{\omega}/\sqrt{h}}{2 - i\hat{\omega}/\sqrt{h}}\right)^{i\sqrt{h}\hat{\omega}} \big(1 + \mathcal{O}(1/h)\big), \tag{5.50}$$

where we rescaled $\hat{\omega} = \sqrt{h}\tilde{\omega} = \omega/\sqrt{h}$. The key aspect of this rescaling is that we can now take the limit $h \to \infty$ in the integrand and then evaluate the integral, which is finite and yields

$$\int_0^\infty d\hat{\omega} \lim_{h \to \infty} \left[\hat{\omega} \coth(\pi \sqrt{h}\hat{\omega}) \left(1 + \frac{\hat{\omega}^2}{4h}\right)^{2h - 1/2} \left(\frac{2 + i\hat{\omega}/\sqrt{h}}{2 - i\hat{\omega}/\sqrt{h}}\right)^{i\sqrt{h}\hat{\omega}}\right] = \int_0^\infty d\hat{\omega} \, \hat{\omega} \, e^{-\hat{\omega}^2/2} = 1. \tag{5.51}$$

Plugging the limit (5.51) into (5.50) then establishes

$$2\pi \Delta \langle H_R \rangle \big|_{h \gg 1} = \sqrt{2\pi h} + \dots, \tag{5.52}$$

where the ellipsis refers to terms that vanish as $h$ tends to infinity.

Consistently with the CFT calculation in section 5.4.4 below, taking into account the first order bulk corrections leads to a cancellation of the $\sqrt{h}$ terms in the full expression for HEE,

$$S = S_{\text{RT}} + 2\pi \Delta \langle H_R \rangle = \frac{c}{3} \ln \frac{2}{z_{\text{cut}}} + 4h + \mathcal{O}(1/c) + \mathcal{O}(1/\sqrt{h}). \tag{5.53}$$

### 5.4.3 Holographic QNEC for half interval

At half-interval and large weight the RT part of the QNEC expression (5.32) expands as

$$S_{\text{RT}}'' + \frac{6}{c}\left(S_{\text{RT}}'\right)^2 = -\frac{c}{24} + h - \frac{h}{4}\sqrt{2\pi h} + \dots, \tag{5.54}$$

where the ellipsis denotes terms that either vanish or grow more slowly than linearly in $h$ in the large weight limit. Note that the last "correction" term in (5.54) actually dominates at large weight, since it grows like $h^{3/2}$.

To compute the corrections to QNEC from bulk entanglement again we first look at the modular Hamiltonian. For this we deform the entangling region away from the half-space, again boosting the interval to an equal-time slice. The integrals appearing in the modular Hamiltonian yield

$$\Delta\langle H_0\rangle = \frac{\sqrt{\pi}\,\Gamma(2h+1)}{\Gamma\left(2h+\frac{1}{2}\right)} - \pi h|\lambda| + \left[\pi^{-3/2}\,h\,\Gamma(2h)\left(\frac{\pi^2 h}{\Gamma\left(2h+\frac{1}{2}\right)} + \frac{8h(h+1)-3}{\Gamma\left(2h+\frac{5}{2}\right)}\right) - \frac{h}{2}\right]\lambda^2 + O\left(\lambda^3\right). \tag{5.55}$$

We arrive at the following result for the right-hand side of QNEC (remembering to take $\lambda/2$ derivatives, and adding the RT and $\mathcal{O}(c)$ part)[6]:

$$S'' + \frac{6}{c}\left(S'\right)^2 = -\frac{c}{24} + \frac{3h}{4} + \frac{\left(32h(h+1) - \pi^2 h(4h+3) - 12\right)\Gamma(2h+1)}{16\pi^{3/2}\Gamma\left(2h+\frac{5}{2}\right)} + \mathcal{O}(\Delta\varphi - \pi). \tag{5.56}$$

The last term in this equation is of subleading order $\sqrt{h}$ and not necessarily accurate in our perturbative approximation. Nevertheless, within the framework of the modular Hamiltonian, it reproduces well the numerical result shown in Fig. 8, where it can be seen that for $h = 1$ we indeed have $S'' + \frac{6}{c}\left(S'\right)^2 \approx -c/24 + 0.717$.

The calculation for QNEC using the Bogoliubov coefficients works analogously to the HEE calculation above, except that we now need to boost the interval to have again a constant time-slice (recall that for QNEC we need to shift one of the endpoints in a lightlike direction). It is not completely clear how to deal with boosts in the Bogoliubov coefficients. Fortunately, for the order in $h$ of interest the result turns out to be insensitive to these details, and all that matters is that we correctly take into account the variation of the (proper) length of the interval itself.[7] We checked this by comparing to the modular Hamiltonian presented above. Both methods lead to the same result for QNEC to linear order in the weight, though they differ to order $\sqrt{h}$.

Our final result is

$$S'' + \frac{6}{c}\left(S'\right)^2\Big|_{\Delta\varphi=\pi} = -\frac{c}{24} + \frac{3h}{4} + \mathcal{O}(\sqrt{h}) + \mathcal{O}(1/c). \tag{5.57}$$

We have neglected second order bulk corrections, which are expected to contribute to (and possibly cancel the) order $\mathcal{O}(\sqrt{h})$. Note that the quadratic term in $S'$ only contributes to subleading $\mathcal{O}(1/c)$ corrections, which we neglect.

Our main conclusion is that QNEC holds, but does not saturate for the half-interval at large $h$.

$$\text{Half-interval:}\qquad 2\pi\langle T_{\pm\pm}\rangle - S'' - \frac{6}{c}\left(S'\right)^2\Big|_{\Delta\varphi=\pi} = \frac{h}{4} + \dots. \tag{5.58}$$

The gap in the QNEC non-saturation (5.58) is given by one quarter of the weight.

---

[6]In principle, due to the absolute value in $\pi h|\lambda|$ in (5.55) the second derivative of the EE is not defined at $\Delta\varphi = \pi$. However, interpreting the QNEC-derivative as a generalized distributional derivative would give rise to a $-2\pi h\delta(\Delta\varphi - \pi)$ function.

[7]These calculations are available at http://quark.itp.tuwien.ac.at/~grumil/QNEClargeh.nb.

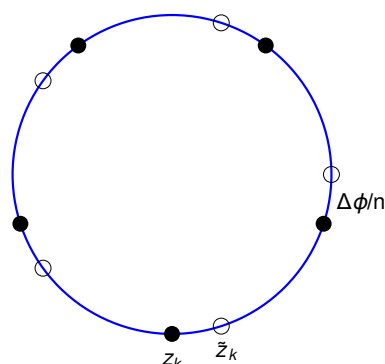

Figure 9: Wick contraction in (5.59) of the 10 operators needed to compute the 5th Renyi entropy [38].

### 5.4.4 CFT analysis of entanglement entropy at large weight

It is illuminating to repeat (part of) the computation above from the CFT side, using the replica trick in the large-$h$ limit. The leading correction to EE is then given by the limit $n \to 1^+$ of the expression for the $n^{\text{th}}$ Renyi entropy [38, 41]

$$\Delta S_n = \frac{1}{1-n} \log \text{Tr} \frac{\rho_A^n}{\rho_{A,vac}^n} = \frac{1}{1-n} \log \left[ e^{-i\Delta\varphi(h-\bar{h})} \left( \frac{2}{n} \sin \frac{\Delta\varphi}{2} \right)^{2n(h+\bar{h})} \left\langle \prod_{k=0}^{n-1} O(\tilde{z}_k) O(z_k) \right\rangle \right], \tag{5.59}$$

where the operators $z_k$ and $\tilde{z}_k$ are located in the complex plane as in Fig. 9. Using large-$c$ factorization the $2n$-point function in (5.59) simplifies to a combinatorial sum of $n$ 2-point functions. The Wick contractions of these 2-point functions lead to

$$\Delta S_n = \frac{1}{1-n} \log \left[ \left( \frac{1}{n} \sin \frac{\Delta\varphi}{2} \right)^{4nh} \text{Hf}(M_{ij}) \right], \tag{5.60}$$

where the Hafnian Hf of the matrix $M$ is defined by

$$\text{Hf}(M) = \frac{1}{2^n n!} \sum_{g \in S_{2n}} \prod_{j=1}^{n} M_{g(2j-1),g(2j)} \tag{5.61}$$

and

$$M_{i,j} = \begin{cases} |\sin(\pi(i-j)/n)|^{-4h} & \text{if } i, j \leq n \\ |\sin(\pi(i-j)/n - \Delta\varphi/(2n))|^{-4h} & \text{if } i \leq n, \quad j > n \\ |\sin(\pi(i-j)/n + \Delta\varphi/(2n))|^{-4h} & \text{if } j \leq n, \quad i > n \\ |\sin(\pi(i-j)/n)|^{-4h} & \text{if } i, j > n. \end{cases} \tag{5.62}$$

The sum in (5.61) goes over all elements $g$ of the permutation group $S_{2n}$, as appropriate for the Wick contraction. The large $h$ limit allows to perturbatively compute the expression for the Hafnian. To see this let us focus on $\Delta\varphi < \pi$. As (5.62) shows the Hafnian in general involves a sum over

$$\frac{1}{n!} \binom{2n}{2} \binom{2n-2}{2} \cdots \binom{2}{2} = \frac{(2n)!}{2^n n!} \tag{5.63}$$

terms. Each term is a product of the form

$$I_n(\alpha_k) \equiv \frac{1}{\sin^{4h} \alpha_1 \sin^{4h} \alpha_2 \cdots \sin^{4h} \alpha_n} = \frac{1}{\sin^{4nh} \alpha_1} \prod_{k=1}^{n} \left( \frac{\sin \alpha_1}{\sin \alpha_k} \right)^{4h}, \tag{5.64}$$

where $\frac{\Delta\varphi}{2n} \leq \alpha_1 \leq \alpha_2 \leq \cdots \leq \alpha_n < \pi/2$.

At large $h$ the sum is dominated by two terms, whereby one contracts two neighbouring operators in Fig. 9. These terms are given by

$$I_n(\alpha_k) = \frac{1}{|\sin\frac{\Delta\varphi}{2n}|^{4nh}} + \frac{\sigma(n)}{|\sin\frac{2\pi-\Delta\varphi}{2n}|^{4nh}} + \text{terms exponentially suppressed in } h. \tag{5.65}$$

In this equation we encounter a subtlety. For the special case of $n = 1$ the Wick contraction has only one term, and both terms in (5.65) are in fact equal and represent this single term. This is reminiscent of the 'replica symmetry breaking' seen in other examples, where the replica symmetry of the original state is not the same as the symmetry of the replicated state [42–44]. We address this issue by introducing the function $\sigma(n) = 0$ for $n = 1$ and $\sigma(n) = 1$ for $n > 1$. This function $\sigma(n)$ is also quite essential for the analytic continuation to $n \to 1$ of the Renyi entropy, since by naively restricting to $n \geq 2$ and analytically continuing one would miss this subtlety and obtain a divergent result for $\text{Tr}\rho_A$ itself. Since EE depends on $\sigma'(1)$ the analytic continuation of this function is not unique [43, 44].[8] In the following we chose $\sigma(n)$ to be parity symmetric around $n = 1$, so that the second term in the end does not affect the analytic continuation $n \to 1$, which indeed gives the result that matches the computation done in the bulk above.

The analytic continuation to obtain the EE is now straightforward, yielding

$$\Delta S = \lim_{n\to 1^+} \frac{4nh}{1-n} \log\left[\frac{\sin\frac{\Delta\varphi}{2}}{n\sin\frac{\Delta\varphi}{2n}}\right] + \text{terms exponentially suppressed in } h$$

$$\simeq 4h\left(1 - \frac{\Delta\varphi}{2}\cot\frac{\Delta\varphi}{2}\right), \tag{5.66}$$

where in the second line we have dropped the exponentially suppressed terms. The analysis for $\Delta\varphi > \pi$ goes along the same lines, and for general $\chi \equiv \pi - \Delta\varphi$ at large $h$ we have the final result

$$\Delta S = 4h\left(1 - \frac{\pi-|\chi|}{2}\cot\frac{\pi-|\chi|}{2}\right). \tag{5.67}$$

It is interesting to compare the result from the replica trick with the holographic computation around $\Delta\varphi = \pi$, where the EE is given by

$$\Delta S = 4h + \pi h|\Delta\varphi - \pi| + h(\Delta\varphi - \pi)^2 + \frac{1}{12}\pi h|\Delta\varphi - \pi|^3 + \frac{1}{12}h(\Delta\varphi - \pi)^4 + O\left((\Delta\varphi - \pi)^5\right). \tag{5.68}$$

In this regime the RT part and the bulk EE can be seen to have three different contributions. The first are terms that cancel when adding up the RT and bulk EE. These cancelling contributions are all of the form $h^{(2n+1)/2}(\Delta\varphi - \pi)^{2k}$, with $n$ and $k$ integers, so that they have fractional powers of $h$. Then the remaining terms of the RT expansion in (5.30) give all the even powers of the expansion:

$$\Delta S_{\text{RT}} \supset 4h + h(\Delta\varphi - \pi)^2 + \frac{1}{12}h(\Delta\varphi - \pi)^4 + \frac{1}{120}h(\Delta\varphi - \pi)^6 + O\left((\Delta\varphi - \pi)^8\right), \tag{5.69}$$

---

[8]Note that there is also an order of limit issue. When taking e.g. the small $\Delta\varphi$ limit first (as in [38]) it is possible to consistently neglect the second term in (5.65) for $n \geq 2$, after which one can take the $n \to 1$ analytic continuation and obtain the correct result. At $\Delta\varphi = \pi$ this alternative strategy unfortunately does not work, even for $h \to \infty$.

whereas the bulk EE from the modular Hamiltonian in (5.35) contains all the odd powers

$$\Delta S_{\text{bulk EE}} \supset \pi h |\Delta\varphi - \pi| + \frac{1}{12}\pi h |\Delta\varphi - \pi|^3 + \frac{1}{120}\pi h |\Delta\varphi - \pi|^5 + O\left((\Delta\varphi - \pi)^7\right). \quad (5.70)$$

Note in particular that the cusp present in the EE at $\Delta\varphi = \pi$, as also found in 5.4.3, can entirely be attributed to the bulk EE.

From this method it is unfortunately not possible to obtain QNEC, since this equation was derived from an equal time entangling region, and our state is not boost invariant. Nevertheless, our CFT result for EE indeed agrees with (5.53) at and around $\Delta\varphi = \pi$, and numerically can be seen to agree at all $\Delta\varphi$ at large $h$, which agrees with the intuition that the large $h$ limit in this case is similar to the small $\Delta\varphi$ expansion pursued in [38].

## 6 Concluding and suggestive remarks

### 6.1 Large weight limit

In section 5.4 we considered HEE and QNEC at half-interval in the limit of large weight, $c \gg h \gg 1$. Here we discuss some general aspects of and open issues with this limit.

While considering the half-interval and taking the large $h$ limit are different procedures, they are related in the following sense. When we take the large $h$ limit for any interval smaller than the half-interval all corrections are suppressed exponentially with the weight, so that QNEC saturates. Thus, in order to get a nontrivial result large $h$ requires to consider the half-interval. On the other hand, when considering the half-interval we do not have a small parameter at our disposal, as required for a perturbative treatment of the excitation density matrix, unless we take the large $h$ limit.

The fact that $1/h$ is a small parameter is necessary for a perturbative treatment at half-interval, but we do not know whether it is sufficient. We collect here the evidence that it might be sufficient. Let us first step back and consider the small-interval limit at large $h$. The first order bulk corrections to HEE calculated in [38] behave like

$$1^{\text{st}} \text{ order:} \qquad \sqrt{2\pi h}\, \sin^{4h}\frac{\Delta\varphi}{2} + \dots, \qquad (6.1)$$

while the second order bulk corrections lead to a term

$$2^{\text{nd}} \text{ order:} \qquad \frac{\sqrt{\pi}}{4\sqrt{h}}\, \sin^{8h}\frac{\Delta\varphi}{2} + \dots. \qquad (6.2)$$

This means that for small interval the prefactor in front of the angular function in the second order expression (6.2) is suppressed by $h$ as compared to the prefactor in front of the angular function in the first order expression (6.1). Of course, at small interval there is an additional exponential suppression coming from the angular functions which no longer applies to the half-interval. However, we expect that the power suppression in $h$ survives as the interval is made larger. We do not know if third order and higher corrections are suppressed by further powers in the weight. Given our results in the previous section we expect this to be the case.

The arguments above are supported by our calculations in section 5.4. We found that taking into account only first order bulk corrections leads to a result (5.53) for HEE that coincides with the exact CFT result (5.66) at large $h$ and for the half-interval, so that the second order contributions that we did not calculate must be suppressed as compared to the first order contributions. Moreover, the CFT result (5.66) shows that the subleading corrections can only appear in exponentially suppressed terms; in particular there are no terms of the form $h^{-n/2}, n \geq 0$. This is quite non-trivial from the bulk viewpoint, and so far we have only a few

indications that it is correct (see section 5.4.2): 1. the terms linear in $h$ come out correctly in the HEE calculation, 2. the $\sqrt{h}$-terms from RT corrections cancel precisely with corresponding terms from first order bulk entanglement corrections, 3. subleading terms to order $1/\sqrt{h}$ remain in the holographic computation, but this is precisely the parametric order at which second order bulk entanglement corrections are expected to kick in, see (6.2) above. It would be interesting to verify holographically the CFT prediction that second order bulk entanglement corrections lead to a cancellation of all terms of order $1/\sqrt{h}$, along the lines of section 5.4.2 and section 5.3 in [38].

Regarding QNEC at half-interval and large $h$, we have only performed the calculation on the gravity side, where again we observed an intriguing precise cancellation of half-integer power terms, $h^{3/2}$, between RT corrections and first order bulk corrections, see section 5.4.3. This suggests that we are on the right track. Apart from pushing this calculation to second order bulk entanglement corrections, it would be of interest to reproduce our holographic computation on the CFT side, essentially by boosting the state dual to the quantum backreacted gravity solution discussed in section 5.1. We leave this to future work.

## 6.2 Quantum equilibrium = QNEC saturation

Motivated by the examples discussed in the previous sections we introduce the notion of quantum equilibrium. We define a state to be in quantum equilibrium if it saturates QNEC for all times and entangling regions. We note that our notion of quantum equilibrium should be distinguished from thermal equilibrium. As the example in section 3.2 shows, it is possible to have a far from thermal equilibrium system that nevertheless always saturates QNEC, and hence relaxation toward thermal equilibrium happens on a path which is in "quantum equilibrium". In the other example we studied, the AdS$_3$ Vaidya quench, the system is out-of-quantum-equilibrium as there is always an entangling region where QNEC does not saturate. The example studied in section 5, in particular the half-interval case of section 5.4, provides another system where QNEC does not saturate, and where the non-saturation happens in a time-independent, static system. The non-saturation in this case is attributed to the "bulk entanglement" in the holographic computation.[9]

For time-dependent states that are not in quantum equilibrium one may define a "quantum relaxation time," which measures how fast the system quantum-equilibrates. It is possible to make a statement about the state approaching quantum equilibrium, for instance at late times: for a given separation of the entangling region $l$ one can define the quantum relaxation time as the smallest time when the normalized QNEC non-saturation is lower than some prescribed (small) value $\epsilon$, e.g. $\epsilon = 1\%$, and remains lower than that value for all future times:

$$\tau_{\text{QEQ}}(l) = \text{minimum, such that } 1 - \frac{S''(l) + \frac{6}{c}\left(S'(l)\right)^2}{2\pi \langle T_{\pm\pm} \rangle}\bigg|_t \leq \epsilon \qquad \forall t \geq \tau_{\text{QEQ}}. \qquad (6.3)$$

For instance, for separation $l = 5.0$ the quantum relaxation time of the Vaidya system with, say, $a = 30$ plotted in the right Fig. 6 can be read off as $\tau_{\text{QEQ}} \approx 2.5$ for reasonably small values of $\epsilon$; this means that for negative times the system is in quantum equilibrium, between $t > 0$ and $t < \tau_{\text{QEQ}}$ the system is out of quantum equilibrium (due to the presence of the Vaidya matter shockwave) and for $t \geq \tau_{\text{QEQ}}$ the system goes back to quantum equilibrium.

Quantum relaxation time depends in general on the size of the entangling region; in practical applications there is probably a reasonable range of choices for the size of this region (related to dimensions of the system) so that the notion of quantum relaxation time becomes

---

[9]Recall that HEE receives contributions from the RT part and the bulk EE. For the half-interval case, the RT part in this case saturates QNEC. It is interesting to check if having a non-zero bulk entanglement always amounts to a non-saturation of QNEC, and hence, within our proposal, out-of-quantum-equilibrium.

meaningful. It is interesting to note that one system could have a range of quantum relaxation times depending on the size of the entangling regions under consideration, so that different scales quantum equilibrate at different times.

It remains to be seen if the notion of quantum equilibrium and quantum relaxation time introduced above is useful for applications. If so, we expect the quantum equilibration time(s) (6.3) to capture essential time scales associated with the dynamics of the underlying quantum system.

## 6.3 Towards operator interpretation of QNEC

Before providing our proposal for an operator interpretation of QNEC we need to make a detour through the gravity side. All Bañados geometries solve the vacuum Einstein equations in three dimensions (with negative cosmological constant), but not all solutions to the Einstein equations in three dimensions are Bañados geometries. The reason for this is that all Bañados geometries obey Brown–Henneaux boundary conditions [16], but other consistent choices of boundary conditions are possible, see [45] and references therein.

In particular, [46] considered boundary conditions leading to near-horizon conserved currents[10] $\mathbf{J}^\pm$ generating two $u(1)$ current algebras

$$[\mathbf{J}_n^\pm, \mathbf{J}_m^\pm] = \frac{c}{12} n \, \delta_{n+m,0} \,, \tag{6.4}$$

where in this section we use bold-face symbols to denote operators. Matching to asymptotic variables these currents uniquely induce spin-2 currents through a twisted Sugawara-construction [46]

$$\mathbf{L}_n^\pm = \frac{6}{c} \sum_p \mathbf{J}_{n-p}^\pm \mathbf{J}_p^\pm + in \mathbf{J}_n^\pm \,, \tag{6.5}$$

where $c$ is the Brown–Henneaux central charge (2.1).[11] The commutation relation between Virasoro generators and current algebra generators compatible with the transformation behavior (6.7) is given by [47]

$$[\mathbf{L}^\pm{}_n, \mathbf{J}^\pm{}_m] = -m \mathbf{J}^\pm{}_{n+m} + i \frac{c}{12} n^2 \, \delta_{n+m,0} \,. \tag{6.6}$$

Let us now come back to (derivatives) of HEE and QNEC. Identifying

$$\frac{dS}{dx_1^\pm} = \sum_n e^{inx^\pm} \langle \mathbf{J}_n^\pm \rangle = J^\pm(x^\pm) \tag{6.7}$$

relates the derivative of HEE with respect to null variations of one of the endpoints to the $u(1)$ currents (6.4). Given the discussion above it is suggestive to lift $S'$ to an operator, in the

---

[10]For states dual to gravity solutions with horizon, like BTZ black holes, the notion of near-horizon current is literally what the name suggests — a current defined through a near-horizon expansion [46,47]. For other Bañados states (like particles or global AdS$_3$) there is no horizon, but the "near-horizon currents" can still be defined and have imaginary expectation values for their zero modes [48].

[11]The above conserved charges are functions on the phase space associated with Bañados geometries (2.3) where each point is specified by $\mathcal{L}^\pm$ functions. On this phase space the currents are also functions $\mathcal{J}^\pm(x^\pm)$, where the identity equivalent to (6.5) becomes $\mathcal{L}^\pm = \frac{6}{c} \mathcal{J}^{\pm 2} + \mathcal{J}^{\pm\prime}$, where $\mathcal{J}$ are the currents associated to the conserved (soft) charges of the near horizon geometry of the dual black hole. Note also that the commutator (6.4) is nothing but the quantized form of the Poisson bracket of currents on this phase space, where the Fourier modes of the functions $\mathcal{J}, \mathcal{L}$ are promoted to quantum operators $\mathbf{J}_n^\pm, \mathbf{L}_n^\pm$ and the phase space to a Hilbert space, see e.g. [27,47,48] and references therein. In (6.5) we have an ordering ambiguity in the $J^2$ term. One may choose normal ordering which leads to a quantum shift of the central charge by 1. Since we work in the large $c$ limit normal ordering does not play a role.

same way we quantized the current modes $J_n$, i.e., one can view the left hand side of (6.7) as derivative of the expectation value of an operator $\mathbf{S}$.

Through the near horizon detour above we have explicitly identified a candidate for the operator version of $S'$, namely the current operator corresponding to the near-horizon current. As a simple sanity check, we verify now that the known transformation behavior of HEE under Penrose–Brown–Henneaux diffeomorphisms [15]

$$\delta_\xi S = -\xi^\mu \, \partial_\mu S + \frac{c}{12} \, \partial_\mu \xi^\mu \tag{6.8}$$

is compatible with the identification (6.7). Differentiating (6.8), say, with respect to $x^+$ and using (6.7) yields

$$\delta_{\xi^+} J^+ = -\xi^+ J^{+\prime} - \xi^{+\prime} J^+ + \frac{c}{12} \, \xi^{+\prime\prime}, \tag{6.9}$$

which is precisely the transformation behavior of a (twisted) $u(1)$ current with (6.4) and (6.6).

The considerations above suggest that the right-hand side of the QNEC inequality (1.1) has an operator interpretation. However, we do not know how the discussion above is modified in the presence of bulk matter in the gravity dual. Given the relation between modular Hamiltonian and EE variations through the first law of EE [49] it is perhaps not surprising that QNEC, which features variations of EE, naturally suggest an operator interpretation. In fact, the analyses and discussions of [38, 50] support the existence of such an operator interpretation. The novel aspect of our discussion is to relate $S'$ to the near horizon currents through (6.7).

## Acknowledgements

We thank Alex Belin for discussions. DG thanks Julian Sonner for drawing his attention to [13] and for discussions. This work was supported by the Austrian Science Fund (FWF), projects P 27182-N27, P 28751-N27 and W 1252-N27. CE was in addition supported by the Delta-Institute for Theoretical Physics (D-ITP) that is funded by the Dutch Ministry of Education, Culture and Science (OCW). DG was supported by the Austrian Science Fund (FWF), projects P 28751, P 30822. WS is supported by VENI grant 680-47-458 from the Netherlands Organisation for Scientific Research (NWO). MMShJ was supported by the grants from ICTP NT-04, INSF junior chair in black hole physics, grant No 950124 and Saramadan grant no. ISEF/M/97219. DG and MMShJ acknowledge an Iran-Austria IMPULSE project grant, supported and run by Khawrizmi University.

## A  Entanglement entropy for two connected heat baths

For the example discussed in section 3.2 one can explicitly work out the expression for the EE (2.9) using the solutions given in (3.7). In particular $\ell^\pm$ (2.10) consists of the multiplication of several theta functions as a function of different combinations of $x_1^\pm \equiv t_1 \pm x_1$ and $x_2^\pm \equiv t_2 \pm x_2$. For all separate combinations EE is an easy analytic expression, which we give in this appendix. We display now for all possible domains the expression obtained for $e^{6 S_{\text{EE}}/c}$.

- Domain $x_{1,2}^\pm > 0$

$$\frac{\sinh\left(\pi T_L(x_1^- - x_2^-)\right)\sinh\left(\pi T_R(x_1^+ - x_2^+)\right)}{\pi^2 \epsilon^2 T_L T_R} \tag{A.1}$$

- Domain $x_1^\pm > 0, x_2^- > 0, x_2^+ < 0$

$$\frac{\sinh\left(\pi T_L(x_1^- - x_2^-)\right)\left(T_L \cosh\left(\pi x_2^+ T_L\right)\sinh\left(\pi x_1^+ T_R\right) - T_R \sinh\left(\pi x_2^+ T_L\right)\cosh\left(\pi x_1^+ T_R\right)\right)}{\pi^2 \epsilon^2 T_L^2 T_R}$$

(A.2)

- Domain $x_1^\pm > 0, x_2^- < 0, x_2^+ > 0$

$$\frac{\sinh\left(\pi T_R(x_1^+ - x_2^+)\right)\left(T_R \sinh\left(\pi x_1^- T_L\right)\cosh\left(\pi x_2^- T_R\right) - T_L \cosh\left(\pi x_1^- T_L\right)\sinh\left(\pi x_2^- T_R\right)\right)}{\pi^2 \epsilon^2 T_L T_R^2}$$

(A.3)

- Domain $x_1^\pm > 0, x_2^\pm < 0$

$$\frac{T_R \sinh\left(\pi x_1^- T_L\right)\cosh\left(\pi x_2^- T_R\right) - T_L \cosh\left(\pi x_1^- T_L\right)\sinh\left(\pi x_2^- T_R\right)}{\pi^2 \epsilon^2 T_L^2 T_R^2} \times$$
$$\times \left(T_L \cosh\left(\pi x_2^+ T_L\right)\sinh\left(\pi x_1^+ T_R\right) - T_R \sinh\left(\pi x_2^+ T_L\right)\cosh\left(\pi x_1^+ T_R\right)\right) \quad \text{(A.4)}$$

- Domain $x_1^- > 0, x_1^+ < 0, x_2^\pm > 0$

$$\frac{\sinh\left(\pi T_L(x_1^- - x_2^-)\right)\left(T_R \sinh\left(\pi x_1^+ T_L\right)\cosh\left(\pi x_2^+ T_R\right) - T_L \cosh\left(\pi x_1^+ T_L\right)\sinh\left(\pi x_2^+ T_R\right)\right)}{\pi^2 \epsilon^2 T_L^2 T_R}$$

(A.5)

- Domain $x_1^- > 0, x_1^+ < 0, x_2^- > 0, x_2^+ < 0$

$$\frac{\sinh\left(\pi T_L(x_1^- - x_2^-)\right)\sinh\left(\pi T_L(x_1^+ - x_2^+)\right)}{\pi^2 \epsilon^2 T_L^2}$$

(A.6)

- Domain $x_1^- > 0, x_1^+ < 0, x_2^- < 0, x_2^+ > 0$

$$\frac{T_L \cosh\left(\pi x_1^- T_L\right)\sinh\left(\pi x_2^- T_R\right) - T_R \sinh\left(\pi x_1^- T_L\right)\cosh\left(\pi x_2^- T_R\right)}{\pi^2 \epsilon^2 T_L^2 T_R^2} \times$$
$$\times \left(T_L \cosh\left(\pi x_1^+ T_L\right)\sinh\left(\pi x_2^+ T_R\right) - T_R \sinh\left(\pi x_1^+ T_L\right)\cosh\left(\pi x_2^+ T_R\right)\right) \quad \text{(A.7)}$$

- Domain $x_1^- > 0, x_1^+ < 0, x_2^\pm < 0$

$$\frac{\sinh\left(\pi T_L(x_1^+ - x_2^+)\right)\left(T_R \sinh\left(\pi x_1^- T_L\right)\cosh\left(\pi x_2^- T_R\right) - T_L \cosh\left(\pi x_1^- T_L\right)\sinh\left(\pi x_2^- T_R\right)\right)}{\pi^2 \epsilon^2 T_L^2 T_R}$$

(A.8)

- Domain $x_1^- < 0, x_1^+ > 0, x_2^\pm > 0$

$$\frac{\sinh\left(\pi T_R(x_1^+ - x_2^+)\right)\left(T_L \cosh\left(\pi x_2^- T_L\right)\sinh\left(\pi x_1^- T_R\right) - T_R \sinh\left(\pi x_2^- T_L\right)\cosh\left(\pi x_1^- T_R\right)\right)}{\pi^2 \epsilon^2 T_L T_R^2}$$

(A.9)

- Domain $x_1^- < 0, x_1^+ > 0, x_2^- > 0, x_2^+ < 0$

$$\frac{T_L \cosh\left(\pi x_2^- T_L\right)\sinh\left(\pi x_1^- T_R\right) - T_R \sinh\left(\pi x_2^- T_L\right)\cosh\left(\pi x_1^- T_R\right)}{\pi^2 \epsilon^2 T_L^2 T_R^2} \times$$
$$\times \left(T_L \cosh\left(\pi x_2^+ T_L\right)\sinh\left(\pi x_1^+ T_R\right) - T_R \sinh\left(\pi x_2^+ T_L\right)\cosh\left(\pi x_1^+ T_R\right)\right) \quad \text{(A.10)}$$

- Domain $x_1^- < 0, x_1^+ > 0, x_2^- < 0, x_2^+ > 0$

$$\frac{\sinh\left(\pi T_R(x_1^- - x_2^-)\right)\sinh\left(\pi T_R(x_1^+ - x_2^+)\right)}{\pi^2\epsilon^2 T_R^2} \tag{A.11}$$

- Domain $x_1^- < 0, x_1^+ > 0, x_2^\pm < 0$

$$\frac{\sinh\left(\pi T_R(x_1^- - x_2^-)\right)\left(T_L\cosh\left(\pi x_2^+ T_L\right)\sinh\left(\pi x_1^+ T_R\right) - T_R\sinh\left(\pi x_2^+ T_L\right)\cosh\left(\pi x_1^+ T_R\right)\right)}{\pi^2\epsilon^2 T_L T_R^2} \tag{A.12}$$

- Domain $x_1^\pm < 0, x_2^\pm > 0$

$$\frac{T_L\cosh\left(\pi x_2^- T_L\right)\sinh\left(\pi x_1^- T_R\right) - T_R\sinh\left(\pi x_2^- T_L\right)\cosh\left(\pi x_1^- T_R\right)}{\pi^2\epsilon^2 T_L^2 T_R^2}\times$$
$$\times\left(T_R\sinh\left(\pi x_1^+ T_L\right)\cosh\left(\pi x_2^+ T_R\right) - T_L\cosh\left(\pi x_1^+ T_L\right)\sinh\left(\pi x_2^+ T_R\right)\right) \tag{A.13}$$

- Domain $x_1^\pm < 0, x_2^- > 0, x_2^+ < 0$

$$\frac{\sinh\left(\pi T_L(x_1^+ - x_2^+)\right)\left(T_L\cosh\left(\pi x_2^- T_L\right)\sinh\left(\pi x_1^- T_R\right) - T_R\sinh\left(\pi x_2^- T_L\right)\cosh\left(\pi x_1^- T_R\right)\right)}{\pi^2\epsilon^2 T_L^2 T_R} \tag{A.14}$$

- Domain $x_1^\pm < 0, x_2^- < 0, x_2^+ > 0$

$$\frac{\sinh\left(\pi T_R(x_1^- - x_2^-)\right)\left(T_R\sinh\left(\pi x_1^+ T_L\right)\cosh\left(\pi x_2^+ T_R\right) - T_L\cosh\left(\pi x_1^+ T_L\right)\sinh\left(\pi x_2^+ T_R\right)\right)}{\pi^2\epsilon^2 T_L T_R^2} \tag{A.15}$$

- Domain $x_{1,2}^\pm < 0$

$$\frac{\sinh\left(\pi T_L(x_1^+ - x_2^+)\right)\sinh\left(\pi T_R(x_1^- - x_2^-)\right)}{\pi^2\epsilon^2 T_L T_R} \tag{A.16}$$

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
