# Peer review of "Quantum Null Energy Condition and its (non)saturation in 2d CFTs"

_SciPost Physics, doi:SciPost Phys. 6, 036 (2019)_

## Round 1 · Referee Report · Anonymous (Referee 1) · 2019-3-4

Strengths

1) Simple and elegant proof of the saturation of the quantum null energy condition for 2d holographic CFTs corresponding to the vacuum solutions of AdS3 gravity.

2) Very detailed (analytical and numerical) analysis of various cases including far-from equilibrium setups, and cases with a bulk scalar and backreactions for which the quantum null energy condition is not saturated.

Weaknesses

I don't quite see any definite weaknesses. Some parts of the paper present somewhat speculative statements, such as the concept of "Quantum equilibrium". So, in principle, I would like to see more supports for such statements, which make the manuscript more valuable. However, these type of statements appear only in the last section, and for the bulk of the paper, the authors are quite concrete. So, I see no problem.

Report

In their paper, the authors studied the quantum null energy condition (QNEC) in 2d holographic CFTs. After providing a simple and elegant proof of the saturation of the QNEC for vacuum solutions of AdS3 Einstein gravity, they studied various cases, including far from equilibrium setups and cases with a bulk scalar field. In the latter case, the QNEC needs not to saturate. They also discussed quantum corrections. I found that the manuscript is interesting and sheds light on the QNEC, and hence recommend the paper to be published once the minor concerns in the following are properly addressed (see below).

Requested changes

1) Throughout the paper, "QNEC" is sometimes used to refer actually the RHS of the QNEC in (1.1). For example, there are sentences like "QNEC diverges". These sentences are somewhat strange, as the QNEC per se cannot diverge. I suggest the authors to find a better way of phrasing these sentences.

2) I don't quite see the merit of the numerics presented in Sec. 3.2. The approximate numerics solution was found by approximating the step function. Just below the numerical paragraph, however the authors presented the exact solution (3.6) and (3.7), without approximating the step function.

3) I don't quite understand what the author would mean by the operator interpretation.The LHS of (1.1) is a number (expectation value), and hence the RHS should also be a number. I also failed to see the connection between the twisted Sugawara construction and the identification (6.7). I would hope that the authors could elaborate on this. I guess that it's perhaps simply a matter of putting brackets < ... > in some expressions like (6.7 - 6.9)?

  • validity: high
  • significance: high
  • originality: high
  • clarity: high
  • formatting: excellent
  • grammar: good

Author:  Daniel Grumiller  on 2019-03-14  [id 466]

(in reply to Report 1 on 2019-03-04)
Category:
reply to objection

We do agree with the referee on the strengths and weaknesses of our work. Regarding the three suggestions for minor changes, we have the following replies.

1) We agree with the referee and have rephrased the few statements where `QNEC' refers only to the right hand side of the QNEC inequality.

2) We have added a brief statement in section 3.2 after Eq. (3.5); while it is true that the analytic discussion of the step-function could be sufficient, we decided to keep the smooth plots in Figure 1 and the corresponding two numerical paragraphs since these plots nicely capture the essence of our statements and corresponding pictures with a theta-function would be harder to decipher.

3) We concede that the referee may have a point and that our presentation was not sufficiently transparent. We have made section 6.3 clearer by introducing bold-faced notation for operators to make it easier to distinguish when we are talking about operators and when we are talking about vacuum expectation values or functions. We adapted the wording after Eq. (6.7) and in footnote 11 accordingly.

---

## Round 2 · Referee Report · Anonymous (Referee 1) · 2019-3-19

Report

The authors revised the manuscript and addressed the concerns raised in my report. I believe that the paper is now ready for publication.

---

## Round 2 · List of Changes

-) corrected the wording of the few remaining instances where 'QNEC' referred to the right hand side of the QNEC-inequality (addresses point 1 by referee)

-) minor text edits in section 3.2 in paragraphs after Eq. (3.5) (addresses point 2 by referee)

-) minor edits in section 6.3 after Eq. (6.7) and in footnote 11 for clarity, in particular introduction of bold faced letters for operators (addresses point 3 by referee)

---

## Editorial Decision

published